# Kidney cytosine methylation changes improve renal function decline estimation in patients with diabetic kidney disease

Caroline Gluck [1,2], Chengxiang Qiu [1,10], Sang Youb Han[3,10], Matthew Palmer[4], Jihwan Park [1],
Yi-An Ko [1,5], Yuting Guan[1], Xin Sheng[1], Robert L. Hanson [6], Jing Huang[7], Yong Chen[7], Ae Seo Deok Park[1],
Maria Concepcion Izquierdo[1], Ioannis Mantzaris[8], Amit Verma [8], James Pullman [9], Hongzhe Li[7] &
Katalin Susztak [1,5]

Epigenetic changes might provide the biological explanation for the long-lasting impact of metabolic alterations of diabetic kidney disease development. Here we examined cytosine methylation of human kidney tubules using Illumina Infinium 450 K arrays from 91 subjects with and without diabetes and varying degrees of kidney disease using a cross-sectional design. We identify cytosine methylation changes associated with kidney structural damage and build a model for kidney function decline. We find that the methylation levels of 65 probes are associated with the degree of kidney fibrosis at genome wide significance. In total 471 probes improve the model for kidney function decline. Methylation probes associated with kidney damage and functional decline enrich on kidney regulatory regions and associate with gene expression changes, including epidermal growth factor (*EGF*). Altogether, our work shows that kidney methylation differences can be detected in patients with diabetic kidney disease and improve kidney function decline models indicating that they are potentially functionally important.

[1] Department of Medicine, Renal Electrolyte and Hypertension Division, University of Pennsylvania, Philadelphia 19104 PA, USA. [2] Department of Pediatrics, Division of Nephrology, The Children's Hospital of Philadelphia, Perelman School of Medicine, University of Pennsylvania, Philadelphia 19104 PA, USA. [3] Division of Nephrology, Department of Internal Medicine, Inje University College of Medicine, Goyang 10380, Korea. [4] Department of Pathology and Laboratory Medicine, Perelman School of Medicine, University of Pennsylvania, Philadelphia 19104 PA, USA. [5] Department of Genetics, University of Pennsylvania, Philadelphia 19104 PA, USA. [6] Diabetes Epidemiology and Clinical Research Section, National Institute of Diabetes and Digestive and Kidney Diseases, Phoenix 86014 AZ, USA. [7] Department of Biostatistics, Epidemiology, and Informatics, Center for Clinical Epidemiology and Biostatistics, School of Medicine, University of Pennsylvania Perelman, Philadelphia 19104 PA, USA. [8] Department of Medicine, Albert Einstein College of Medicine, Bronx 10461 NY, USA. [9] Department of Pathology Montefiore Medical Center, Bronx 10467 NY, USA. [10] These authors contributed equally: Chengxiang Qiu, Sang Youb Han. Correspondence and requests for materials should be addressed to K.S. (email: ksusztak@pennmedicine.upenn.edu)

Diabetes mellitus (DM) affects upwards of half a billion people worldwide and can lead to kidney complications, diabetic kidney disease (DKD), in ~40% of patients with DM[1,2]. The presence of kidney disease can explain most excess mortality associated with diabetes and DKD is the leading cause of end stage kidney disease worldwide[1].

Despite the high clinical need and intense efforts, our understanding of DKD remains limited. Several factors explain this critical information gap. Animal models for diabetes do not develop renal complications, which limits their utility in studying the pathogenesis of DKD and in testing new drugs to treat DKD[3]. In addition, large-scale genetic efforts identified less than a handful of loci that reached genome wide significance for diabetic kidney disease, despite the strong heritability of DKD[4–9]. Finally, while there is increased risk of kidney disease development in patients with poor glycemic control, recent large interventional studies failed to show survival or renal benefit following normalization of blood glucose levels in patients with established diabetes[10–12]. These observations suggest that mechanisms outside of "traditional" genetic variants might be responsible for disease development.

Several groups have recently proposed that metabolic or developmental programming might play an important role in DKD development[13–18]. In the DCCT trial, patients initially assigned to the conventional glucose control group had an increased rate of DKD even after 25 years of strict glucose control, a phenomenon called metabolic memory[19–23]. Similarly, studies show consistent association between intrauterine nutritional deprivation and development of salt sensitive hypertension and kidney disease[24–30]. Epigenetic alterations could provide an explanation for these clinical observations. The epigenome is under the environmental influence and epigenetic changes are maintained during cell division. Therefore, metabolically driven epigenetic changes in the kidney could potentially explain the clinical observational data such as metabolic memory and developmental programming.

Several studies have tried to address the presence and role of epigenetic changes in DKD subjects. Changes in both histone and methylation patterns have been described when blood samples from patients with DM are compared to those with DKD[22,31–35]. While these early reports are interesting, most of these studies used small cohorts and most of the published changes failed to pass the genome wide statistical significance threshold. Furthermore, as epigenetic changes are cell type specific, it is not clear whether changes observed in blood samples correlate with kidney specific epigenetic differences. Our group has previously analyzed genome wide cytosine methylation changes in genomic DNA samples obtained from microdissected kidney samples of patients with mixed cohort of CKD[36], hence the existence and role of epigenetic changes in DKD remains a critically important yet unanswered question.

While kidney function predictably declines with age, defining the speed of decline and detection of patients who will require renal replacement therapy is a critically important issue[37,38]. Patients with fast functional decline must be prioritized for intervention. Recent studies show that that baseline glomerular filtration rate (GFR), albuminuria and blood biochemical parameters can fairly accurately predict kidney function decline[39]. Additional biomarkers in the blood or urine have been identified recently that also predict renal function decline, however none has been shown to outperform the baseline clinical parameters. Epigenetic changes are the footprint of prior environmental alterations, but they are stable and inherited during cell division, therefore they could be ideal disease biomarkers.

The goal of the current project is to define genome wide cytosine methylation differences as measured by the Illumina Infinium 450k array in microdissected human kidney tubule epithelial cells of patients with diabetes and kidney disease. We integrate methylation changes with cell-type specific regulatory maps and gene expression changes to understand whether or not the observed changes are functionally important. Furthermore, we apply machine-learning algorithms to clinical and pathological descriptors to create renal function decline models and find that cytosine methylation levels can improve current models of renal function decline.

## Results

**Methylation probes associate with degree of interstitial fibrosis**. To describe epigenetic changes in diabetic kidney disease we analyzed microdissected human kidney tubule samples. As cytosine methylation is cell type specific, it is essential to study disease relevant cell types[40]. Our primary cohort included patients with and without diabetes and hypertension and varying degrees of diabetic kidney disease (DKD) as well as controls for comparison. The study used a cross sectional design and primary cohort sample size was 91. Demographics, clinical and histopathological analysis is shown on Tables 1 and 2. Histopathological analyses and quantitative scoring of tubulointerstitial fibrosis were performed by a renal pathologist. Principal component analysis for our primary cohort methylation data is available in Supplementary Figure 1.

Diabetic kidney disease has multiple manifestations, such as structure changes, mesangial expansion, glomerulosclerosis and tubulointerstitial fibrosis, in addition to functional changes of albuminuria and kidney function (eGFR) changes. The clinical definition of DKD includes albuminuria and GFR decline, however recent studies indicate a relatively poor correlation between these two manifestations. Structural changes are considered to be the gold standard to define DKD, therefore we used renal histological changes (mesangial expansion, fibrosis) to define DKD. The clinical (eGFR) and structural (fibrosis) descriptors showed a good, but not perfect correlation (Supplementary Table 1) and therefore they were analyzed independently. Furthermore, disease presentations follow a continuous pattern and this pattern was reflected in the analysis. Rather than using DKD as an outcome, we set to identify methylation changes associated with structural manifestation of DKD (i.e., tubulointerstitial fibrosis) or kidney function level (eGFR) using a linear regression model adjusted for key variables including age, sex, race, diabetes, hypertension, batch effect, bisulfite conversion efficiency, and degree of lymphocytic infiltrate on histology. To identify methylation changes that are not a consequence of genetic variation we have applied the Gap Hunter method and filtered out a large number of probes that were in the vicinity of regions with nucleotide variation. We found that the methylation level of 203 CpG probes significantly correlated with the degree of interstitial fibrosis using a linear regression with False Discovery Rate (FDR) adjusted $p$-value (FDR < 0.05) to determine genomewide significance (Fig. 1a). Probes that showed significant methylation differences were distributed evenly across the genome (Fig. 1a). A similar number of probes showed lower methylation levels associated with interstitial fibrosis as showed higher methylation levels (Fig. 1b). The top probe that associated with kidney function level (eGFR) had a $p$-value of 1.02e−6 using linear regression (Supplementary Fig. 2).

**Validation of fibrosis-associated methylation probes**. We sought to validate the methylation changes associated with

### Table 1 Demographic and clinical characteristics of primary cohort

|  | Primary cohort |
| --- | --- |
| Subjects (n) | N = 91 |
| Baseline eGFR (ml/min per 1.73 m$^2$) | 68.2 (26.0) |
| Female | 43 (47%) |
| Age | 63.5 (11.5) |
| Race |  |
| Asian | 3 (3%) |
| Caucasian | 19 (21%) |
| African American | 32 (35%) |
| Hispanic | 8 (9%) |
| Multiracial | 14 (15%) |
| Unknown | 15 (16%) |
| Diabetes | 41 (45%) |
| Hemoglobin A1C (for DM) | 6.7 (1.3) |
| Hypertension | 64 (71%) |
| MAP | 93.3 (11.9) |
| Proteinuria: dipstick (0–5) | 1 (1.5) |
| BMI (kg/m$^2$) | 30.5 (9.3) |
| Subjects with longitudinal eGFR data (n) | N = 69 |
| Time span (years) | 2.4 (1.5) |
| Unadjusted GFR Slope (ml/min per 1.73m$^2$ per year) | −5.96 (5.80) |
| Adjusted GFR Slope (ml/min per 1.73 m$^2$ per year) | −4.20 (1.34) |

Data are mean (SD) or n (%)

### Table 2 Histological characteristics of primary cohort

|  | Primary cohort |
| --- | --- |
| n | 84 |
| Hypoperfused glomeruli (0–3) | 0.52 (0.57) |
| Glomerular wall thickening (0–3) | 0.25 (0.66) |
| Mesangial matrix (0–3) | 0.49 (0.86) |
| Mesangial cellularity (0–3) | 0.35 (0.78) |
| KW nodule (0–1) | 0.07 (0.26) |
| Pericapsular fibrosis (0–2) | 0.68 (0.75) |
| Globally sclerotic glomeruli (%) | 10.41 (15.66) |
| Segmentally sclerotic glomeruli (%) | 0.48 (1.49) |
| Tubular atrophy (%) | 12.52 (21.03) |
| Acute tubular injury (%) | 1.27 (5.05) |
| Tubules reabsorption (0–3) | 0.21 (0.46) |
| Interstitial fibrosis (%) | 12.21 (18.59) |
| Plasmacytic infiltrate (0–3) | 0.31 (0.56) |
| Lymphocytic infiltrate (0–3) | 0.78 (0.75) |
| Eosinophilic infiltrate (0–3) | 0.16 (0.43) |
| Vessel medial thickening (0–3) | 0.07 (0.26) |
| Vessel intimal fibrosis (0–3) | 1.23 (0.89) |
| Vessel arteriolar hyalinosis (0–3) | 0.43 (0.78) |

Data are mean (SD)

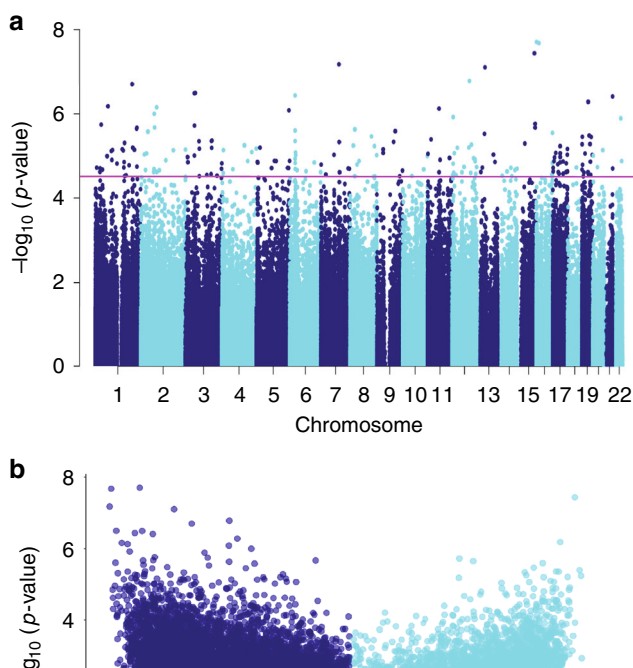

**Fig. 1** Association between cytosine methylation changes and interstitial fibrosis. **a** Manhattan plot of Interstitial Fibrosis associated methylation changes. The x-axis represents the genomic location of the probe, while the y-axis is the negative base 10 log of the p-value. (The association between the methylation level of 321,473 probes and kidney fibrosis was studied using a linear regression models adjusted for age, gender, race, diabetes, hypertension, batch, bisulfite conversion, and degree of lymphocytic infiltrate on histology. The threshold for genome-wide significance was set at p-value FDR < 0.05 as indicated by the horizontal magenta line). **b** Volcano plot depicting the association between Interstitial Fibrosis and methylation changes. The x-axis represents the Pearson correlation coefficient of each probe with Interstitial Fibrosis. The y-axis is the negative base 10 log of the p-value each probe associated with Interstitial Fibrosis

fibrosis, as an important manifestation of DKD. There is a significant overlap in fibrotic changes in diabetic and hypertensive kidney disease therefore we used an independent cohort of 85 samples from patients with mixed diabetic and hypertensive chronic kidney disease (CKD), as well as healthy controls. Demographics, clinical and histopathological descriptors of the validation dataset can be found under Supplementary Tables 2 and 3. Principal component analysis for our replication kidney cohort methylation data is available in Supplementary Figure 3. Again, we used a linear regression analysis while adjusting for age,

sex, race, diabetes, hypertension, batch effect, bisulfite conversion efficiency, and degree of lymphocytic infiltrate on histology. Again, probes potentially influenced by genotypic variation were filtered out using the fairly conservative Gap Hunter method. Of the 203 CpG probes identified in the primary dataset, we replicated the association (p-value < 0.05) for 65 CpG probes using linear regression with directional consistency of the methylation change. In the combined cohort, methylation level of all 65 CpG probes were associated with interstitial fibrosis using linear regression and stringent significance criteria for multiple comparisons, FDR corrected p-value < 0.05, corresponding to nominal p-value < 3.23E−05 (Fig. 2). In summary, methylation level at 65 CpG probes were associated with interstitial fibrosis (independent of CKD etiology) and replicated in an independent cohort using strict statistical correction for multiple comparisons (Supplementary Data 1). Gene ontology and Ingenuity Pathway Analysis (IPA) for replicated methylation probes associated with degree of interstitial fibrosis is shown in Supplementary Table 4 and Supplementary Figure 4, respectively, and included genes associated

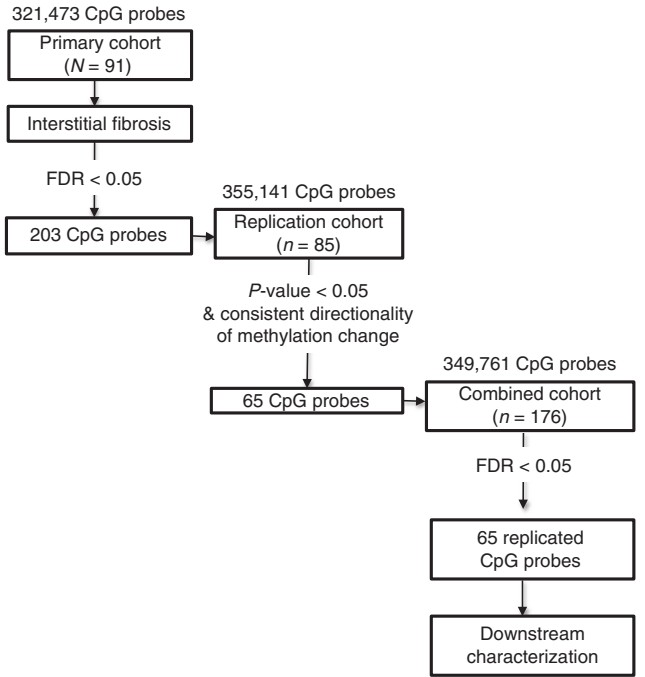

**Fig. 2** Replication of the association between cytosine methylation and intersistial fibrosis Of the 203 probes significantly associated with interstitial fibrosis (FDR < 0.05) in the primary dataset, 174 probes were assessed in the replication data set. 29 probes were excluded due to poor data quality. In total 65 of 174 probes were associated with interstitial fibrosis in the replication data set (p-value < 0.05). All probes had directional consistency of methylation change. The primary data set and replication data set were then combined to reassess for probes significantly associated with interstitial fibrosis (FDR < 0.05). All associations were determined by linear regression models adjusted for age, gender, race, diabetes, hypertension, batch, bisulfite conversion, and degree of lymphocytic infiltrate on histology. Overall, 65 methylation probes associated with interstitial fibrosis were replicated

with cell adhesion, localization, cell death and survival, and connective tissue development and function.

**Fibrosis-associated probes on gene regulatory regions**. Next, we wanted to understand whether methylation changes observed in kidney tubule cells could be functionally important. This follow-up analysis was performed using the 65 probes for which methylation changes were replicated in an independent cohort. Methylation changes that are localized to regions where transcription factors can bind are more likely to be functionally important as they can alter transcriptional accessibility and downstream gene expression levels. We took advantage of the fact that regulatory regions are enriched for specific histone tail modifications to define transcriptionally active regions in the adult human kidney. Healthy human adult kidney ChIP-Seq data was obtained through the Roadmap Epigenomic Project. Using the ChromHMM algorithm we have integrated signals for H3K4me1, H3K4me3, H3K27ac, H3K36me3, H3K9ac, and H3K9me3 data to identify promoter, enhancer and other regulatory regions in the human kidney. We mapped the location of the 65 CpG probes for which methylation level was associated with interstitial fibrosis. As compared with all probes of the HumanMethylation450k array used in the primary analysis (n = 321,473), the 65 CpG probes associated with interstitial fibrosis were enriched in kidney enhancer regions OR 3.49 (95% CI 1.53–7.04, p-value = 0.002) using a two-sided fisher exact

test (Fig. 3a) indicating that are likely to be functionally important. Tissue specific enrichment for these 65 CpG probes is shown in Supplementary Fig. 5. Compared to a random selection of probes our set of 65 probes (that are associated with DKD and fibrosis) showed a 4.5-fold enrichment to be localized to a kidney enhancer region, suggesting their functional importance in the kidney.

**Probes in regulatory regions associate with gene expression**. In order to further support the functional importance of the methylation of the 65 CpG probes in kidney disease development, we correlated methylation changes with gene expression levels analyzed in the same kidney tubule samples. Not all samples had available Affymetrix gene expression data and the demographics and clinical description of this sub-cohort is available in Supplementary Table 5. We correlated CpG probe methylation level change with nearby (500 kb) gene expression changes. We used random permutation method[41] to determine significance between methylation and gene expression and used a cutoff p-value = 8e −5. Using this p-value, we had > 80% power to detect an effect size of beta > 0.5075. The analysis identified 1791 significant methylation-expression correlation pairs (Supplementary Data 2). Since we were interested in methylation changes that are functional in kidney cells, we narrowed our list by looking only at CpG probes located in active kidney regulatory (promoter, enhancer, transcribed) regions. Probes that were significantly associated with interstitial fibrosis and confirmed independent replication (n = 65) were mapped to identify the narrowest set of probes with the highest probability to play a functional role. Five probes were identified that passed these criteria and their association with fibrosis, gene expression and genomic location is shown in Table 3. For top probe, cg20597486, we completed experiments to validate the methylation changes as measured by the Illumina Infinium 450 K arrays (Supplementary Figure 6). Correlations between all top probes associated with degree of interstitial fibrosis and nearby (Cis) gene expression changes is shown in Supplementary Figure 7. Figure 3 further describes one of the top probes: cg20597486. Cg20597486 methylation level was significantly associated with interstitial fibrosis. This probe was located in a kidney promoter region, as shown in Fig. 3b locus zoom[42] indicating that it is likely to be functionally important. While the probe correlated with gene expression changes of multiple nearby genes, it is closest to IFI16 transcription start site (TSS), therefore, this gene is most likely to be influenced by methylation changes at this location. For probes that are further away from the gene TSS, functional link would require further experimental confirmation. Probe methylation level correlated with IFI16 (Gamma-interferon-inducible protein) transcript level in microdissected human kidney tubule samples. These results highlight that decrease of the methylation at cg20597486 is likely to be important for kidney disease, as it is associated with an increase in IFI16 expression.

**Building a model for kidney function decline**. In order to further elucidate methylation changes with increased likelihood of being causal for kidney fibrosis development, we examined probes whose methylation were associated with the rate of future kidney function decline. Multiple factors have already been identified that can predict kidney function decline. Our goal was to understand whether or not cytosine methylation changes independently associate with functional decline (i.e., they are less likely to be a proxy for a known clinical variable). In order to do this, we first defined the association between clinical variables and CKD progression. This analysis was performed in the primary cohort subset with longitudinal eGFR measurements (n = 69) (Clinical information is available Supplementary Table 5). CKD

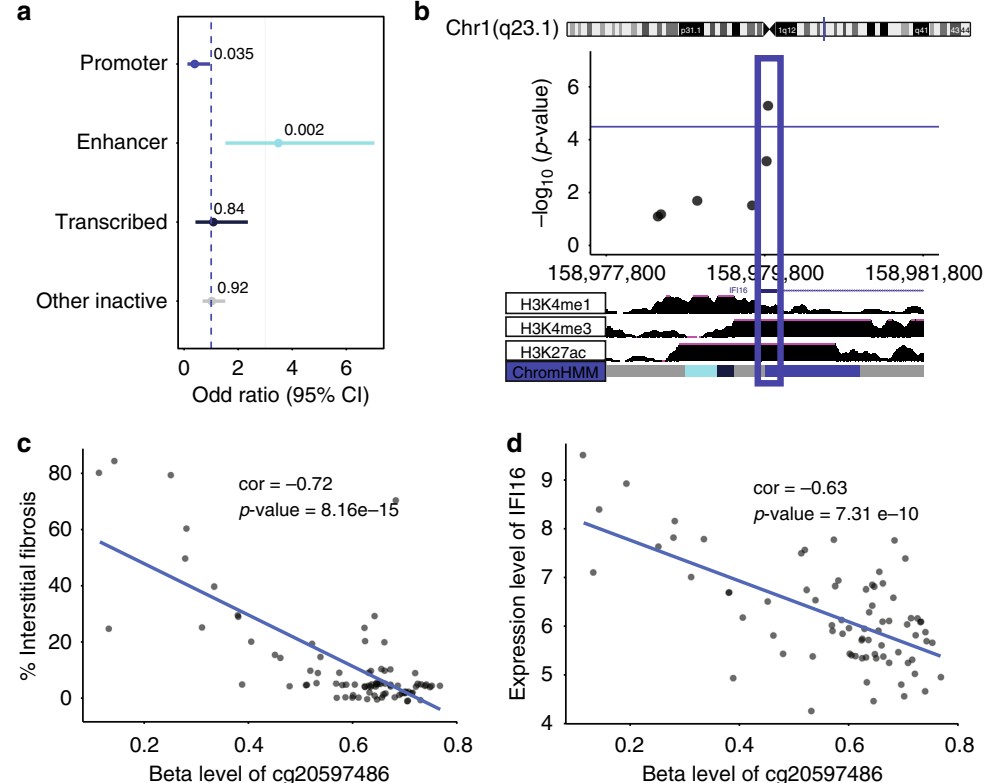

**Fig. 3** Functional annotation of methylation probes associated with interstitial fibrosis. **a** Probes significantly associated with fibrosis were annotated by their regulatory region location. Odds ratios and significance was calculated based on all array probes used in the regression analysis ($n = 321,473$). Probe locations where methylation level is associated with fibrosis were enriched on kidney enhancer regions OR $= 3.49$ (95%CI 1.53–7.04, $p$-value $= 0.002$) using a two-sided fisher exact test. Blue dashed line indicates OR $= 1$. **b** Locus zoom for one top replicated methylation probe (cg20597486) associated with interstitial fibrosis located in kidney promoter region. Chromosome and track images are from UCSC genome browser build hg19 (http://genome.ucsc.edu). **c** Methylation probe (cg20597486) associated with interstitial fibrosis in the primary data set using Pearson correlation test. Unadjusted correlation = -0.72 ($p$-value $= 8.16$ e−15). **d** Methylation probe (cg20597486) associated with IFI16 gene expression using Pearson correlation test. Unadjusted correlation = -0.63 ($p$-value $= 7.31$ e−10)

**Table 3 Top replicated probes associated with interstitial fibrosis and gene expression changes**

| Methylation probe | Primary cohort ($n = 91$) Beta[a] (probe-fibrosis) | Primary cohort ($n = 91$) P-value[a] (probe-fibrosis) | Replication cohort ($n = 85$) P-value[a] (probe-fibrosis) | Position | Kidney regulatory region determined by histone annotation | Cis-gene(s) with expression changes | Distance (bp) from methylation probe to TSS of cis-gene(s) | Beta[c] (probe-cis-gene) | P-value[c] (probe-cis-gene) |
|---|---|---|---|---|---|---|---|---|---|
| cg20597486 | −0.0295 | 4.89e−06 | 2.93e−08 | Chr1: 158979841 | Promoter | AIM2 | 137045 | −0.5587 | 1.23E−07 |
| | | | | | | FCER1A | 279663 | −0.6282 | 7.25E−10 |
| | | | | | | MNDA | 178734 | −0.6630 | 4.58E−11 |
| | | | | | | IFI16[b] | 10083 | −0.6333 | 6.00E−10 |
| cg10512292 | −0.0162 | 1.52e−05 | 4.33e−05 | Chr12: 4378267 | Promoter | CCND2 | 4671 | −0.5151 | 2.47E−06 |
| cg21439672 | −0.0159 | 2.51e−05 | 0.00144 | Chr12: 7260546 | Transcribed | C1S | 164195 | −0.5974 | 4.28E−09 |
| | | | | | | C1R[b] | 15343 | −0.5127 | 4.22E−06 |
| cg05839365 | −0.0104 | 1.83e−05 | 0.00234 | Chr12: 133166658 | Inactive | PXMP2 | 97534 | 0.5254 | 5.35E−07 |
| cg26429629 | −0.0273 | 1.92e−05 | 0.00052 | Chr5: 169758391 | Inactive | LCP2 | 33160 | −0.5093 | 1.95E−06 |

[a] Association determined by linear regression models adjusted for age, gender, race, diabetes, hypertension, batch, bisulfite conversion, and degree of lymphocytic infiltrate on histology
[b] Nearest gene to methylation probe
[c] Association determined by linear regression. Significance determined by random permutation method (cutoff $p$-value $= 8e−5$)

progression was defined by eGFR slope, which was adjusted to account for random variation as well as non-normality (Supplementary Figure 8). Low baseline eGFR and higher interstitial fibrosis was associated with faster eGFR decline (more negative adjusted eGFR slope) as has been described previously (Supplementary Table 6, Supplementary Figure 9)[39,43]. We started with all available clinical and histological variables that correlated with adjusted eGFR slope on univariate analysis ($p$-value $< 0.05$) using

a two-sided Pearson correlation test (Supplementary Table 7). We then selected variables using a machine learning regression analysis method, least absolute shrinkage and selection operator (LASSO), in order to improve model accuracy and reduce model overfitting[44]. In our dataset, the following parameters showed significant (independent) association with the rate kidney function decline: baseline eGFR, diabetes status, and age. These parameters overlap with the published variables known to predict

kidney disease, such as eGFR, age and albuminuria[39] validating our dataset (Supplementary Table 8).

While hypertension has been shown to predict CKD progression in some cohorts[45–47], we examined if the addition of hypertension to our CKD model would alter our outcome. In our data, only the diagnosis of HTN significantly correlated with adjusted eGFR slope on univariate analysis using a two-sided Pearson correlation test (beta = −0.29 $p$-value = 0.017); systolic blood pressure and mean arterial pressure (MAP) were not (beta = −0.07 $p$-value = 0.22; and beta = 0.13 p-value 0.32, respectively) (Supplementary Table 7). Our data is consistent with results of large collaborative cohorts that created kidney function decline prediction models, such as Tangri et al.[4]. These predictive models did not include hypertension or blood pressure diagnosis for functional decline prediction yet were able to reach a C-statistics score of around 0.9[39,48].

**Gene expression does not improve CKD progression model.** Once we confirmed the clinical parameters that predicted functional decline we added genome wide gene expression changes to the linear regression model. All linear regression models were run as a weighted regression whereby subjects with increased variability to their rate of decline were weighted less. Gene expression levels failed to significantly improve our precision to predict kidney function decline at genome wide significance (FDR < 0.05) using a weighted linear regression analysis. Our analysis indicates that failure to improve prediction was not due to a reduced power as the gene expression dataset was smaller ($n = 58$) than the methylation dataset.

**Cytosine methylation improves CKD progression model.** Next, we tested whether epigenetic changes can improve kidney function decline prediction. We individually added CpG probe methylation level to the base CKD progression model. The methylation level of 471 probes was significantly associated with renal function decline (adjusted eGFR slope) at genome wide significance (FDR < 0.05) and improved model fit as measured by model AIC (AIC < 206) (Supplementary Data 3) using a weighted linear regression analysis. Gene ontology annotation for methylation probes that improved our CKD progression model is shown in Supplementary Table 9 and Ingenuity Pathway Analysis in Supplementary Figure 10. We found only minimal overlap between methylation probes that improved kidney function decline and those that correlated with fibrosis (only 2 probes of 65) (Supplementary Table 8). If we force hypertension into our model, we find that 67 of the 471 methylation probes still significantly improve the model (FDR < 0.05) (Supplementary Data 4) using a weighted linear regression analysis. Our results show that kidney tubule methylation changes do improve renal function decline models beyond the already established parameters.

**Top progression-associated probes in gene regulatory regions.** To narrow the probes that not only improved kidney disease prediction, but are likely to be functionally important we examined the functional annotation of the probe location and their association with gene expression. We found that probes that improved kidney disease progression model were more likely to be on kidney specific enhancer regions OR = 2.51 (95% CI 1.82–3.40, $p$-value = 1.04e−7) using a two-sided fisher exact test (Fig. 4a). 131 CpG probes that improve CKD progression models were located in active kidney regulatory regions (promoter, enhancer, transcribed regions). Five CpG probes that significantly improved the CKD progression base model also correlated with nearby gene expression changes (Table 4 and

Supplementary Data 5); one CpG probe was located in an active kidney regulatory region (Table 5). This probe, cg24818418, is shown as an example (Fig. 4b, c). The degree of cg24818418 methylation significantly improved the CKD progression model fit. This probe was located in a kidney promoter region and associated with epidermal growth factor (*EGF*) transcript level. Another top probe, cg21048700, that improved the CKD progression model was associated with changes in *COL3A1* (type III collagen) (Table 5). While this probe was not located in an active regulatory region, when whole kidney samples were analyzed, this probe mapped to an enhancer region in fibroblasts. Correlations between all top probe methylation levels associated with nearby (Cis) gene expression changes is shown in Supplementary Fig. 11. Finally, methylation probes that significantly improve the model of kidney function decline ($n = 471$ CpG probes) were localized on tissue specific enhancer regions (Fig. 4d).

**Validation of top progression-associated methylation probes.** Methylation changes that predict kidney function decline and improve upon existing prediction models are of critical importance therefore we set to validate these results in an independent cohort. Unfortunately, we did not have additional methylation data obtained from human kidney tubule samples with longitudinal eGFR data. On the other hand, we had access to an independent genome wide methylation dataset of peripheral blood mononuclear cells from a well phenotyped cohort of American Indians with detailed longitudinal follow-up that included measurement of eGFR in examinations subsequent to the sample in which methylation was measured. This cohort was also analyzed by Illumina 450k arrays and we ensured that data processing was performed using a similar pipeline (see Methods). Principal component analysis for this cohort is available in Supplementary Fig. 12. Cytosine methylation changes associated with adjusted eGFR slope were determined by a weighted linear regression analysis and controlled for variables such as age, sex, duration of diabetes, mean blood pressure, hemoglobin A1c, batch effect, bisulfite conversion efficiency as well as distribution of cell types. The methylation levels of 25 CpG probes that improved our CKD progression model fit in the kidney also were significantly correlated with kidney function decline in this independent cohort that analyzed blood samples ($p$-value < 0.05 with consistent direction of methylation change) using a weighted linear regression analysis (Fig. 5).

Finally, we compared our results with those recently published by Chen et al. that reported changes in the methylome of blood samples of 63 patients with type1 diabetes form the DCCT/EDIC cohort[22]. We identified some consistencies in the results as shown in Supplementary Data 6, even though we analyzed different cell types (blood vs. kidney), different types of diabetes (DCCT type1 vs. ours type2), and different analytical methods (use of Gap Hunter) and different outcomes (progressive retinopathy/nephropathy vs degree of interstitial fibrosis and eGFR decline). In summary, methylation changes associated with kidney fibrosis can be replicated in other kidney cohort samples and to lesser degree in blood samples.

Overall, DNA cytosine methylation levels of the renal tubules can improve kidney function decline estimations and we have identified specific methylation changes that are associated with kidney function decline.

**Discussion**
To our knowledge this is the first study to describe cytosine methylation differences in kidney tubule samples of patients with DKD. Earlier reported studies used blood samples or analyzed

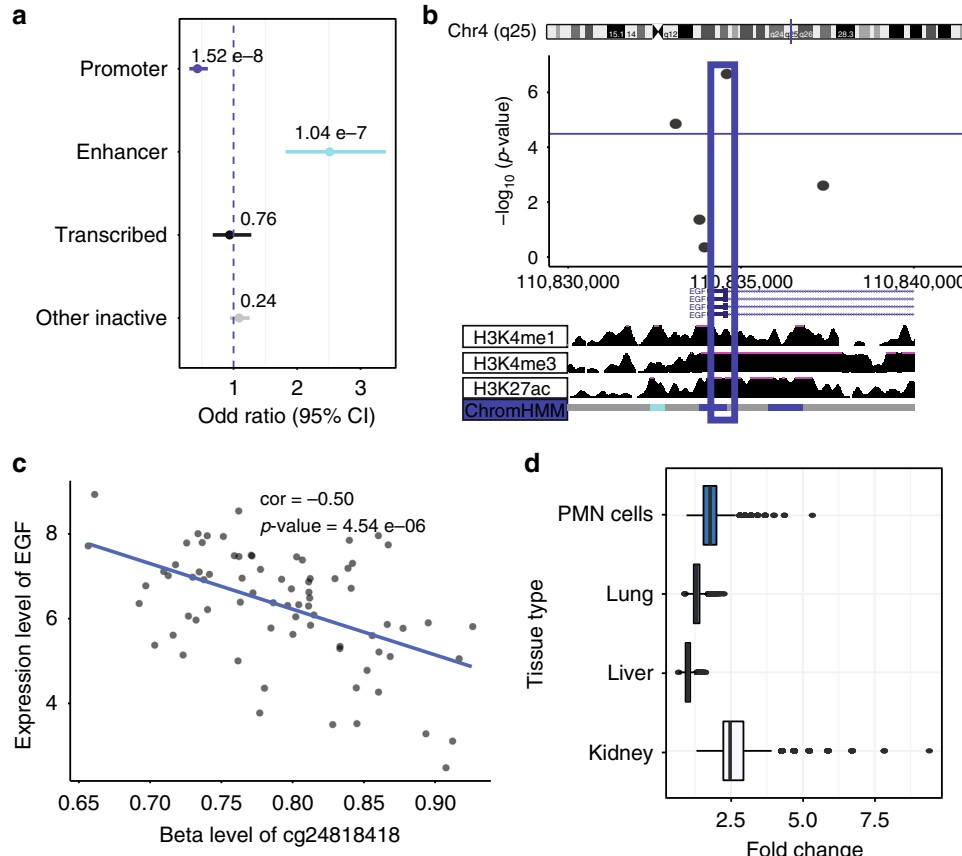

**Fig. 4** Functional annotation of probes associated with kidney function decline. Methylation probes that significantly improve weighted linear regression model of kidney function decline (FDR < 0.05) (n = 471 CpG probes) were annotated by their regulatory region location. **a** Odds ratios and significance was calculated based on all array probes used in the regression analysis (n = 321,473). Probe locations where methylation level improved the baseline kidney function decline model are enriched on kidney enhancer regions OR = 2.51 (95% CI 1.82–3.40, p-value = 1.04 e−7) using a two-sided fisher exact test. Blue dashed line indicates OR = 1. **b** Locus zoom for one CpG probe (cg24818418) that improved the baseline kidney function decline model located in kidney promoter region. Chromosome and track images are from UCSC genome browser build hg19 (http://genome.ucsc.edu). **c** Methylation probe (cg24818418) associated with EGF gene expression level using Pearson correlation test. Unadjusted correlation = -0.50 (p-value = 4.54 e−06). **d** Methylation probes that significantly improve model of kidney function decline (FDR < 0.05) (n = 471 CpG probes) were annotated to tissue specific enhancer region. Fold change compared observed number of significant probes located in each tissue specific enhancer region to the distribution of random 471 probes selected 10,000 times from the background of all array probes used in the regression analysis (n = 321,473). Median fold change for kidney enhancer was 2.47. Median fold change for liver enhancer was 1.00. Median fold change for lung was 1.30. Median fold change for polymorphonuclear (PMN) cells was 1.78. (Center line, median fold change; box limits, upper and lower quartiles; whiskers, 1.5× interquartile range)

### Table 4 Top probes that improve progression model and are associated with gene expression changes

| Methylation probe | AIC[a] | Beta[a] probe-model | P-value[a] (FDR) probe-model | Position | Kidney regulatory region determined by histone annotation | Cis-gene(s) with expression changes | Distance (bp) from methylation probe to TSS of cis-gene (s) | Beta[b] probe-cis-gene | P-value[b] probe-cis-gene |
|---|---|---|---|---|---|---|---|---|---|
| cg09914444 | 190.325 | −0.6040 | 4.84E−05 (0.0459) | Chr1: 46972183 | Other inactive | CYP4A11 | 434973 | 0.5574 | 3.26E−07 |
| cg19506253 | 189.601 | −0.8107 | 3.56E−05 (0.0435) | Chr2:158301839 | Other inactive | CYTIP | 43634 | −0.5077 | 4.67E−06 |
| cg21048700 | 191.121 | −1.2075 | 6.79E−05 (0.0494) | Chr2: 189762162 | Other inactive | COL3A1 | 76884 | −0.5267 | 1.34E−07 |
| cg24818418 | 177.15 | 1.2355 | 1.92E−07 (0.0305) | Chr4: 110834590 | Promoter | EGF | 550 | −0.5156 | 1.30E−06 |
| cg27374758 | 184.453 | 1.0406 | 4.07E−06 (0.0305) | Chr3: 121675259 | Other inactive | HCLS1 | 295485 | 0.5201 | 7.06E−07 |

[a] Model is a weighted linear regression model of adjusted eGFR slope (weight = inverse variance of adjusted eGFR slope). Base model includes variables: baseline eGFR, Diabetes, and Age (base model AIC = 206). When methylation level of probe is added to base model, the following variables are also added: methylation batch, and bisulfite conversion efficiency.
[b] Association determined by linear regression. Significance determined by random permutation method (cutoff p-value = 8e−5)

non-diabetic kidney disease[22,31–35,49]. Most prior studies used modest sample sizes and most of these results failed to pass genome wide significance levels. Changes observed in renal tubule cells are robust compared to results published for blood samples

for similar sample size[22,31–34,49]. DKD has multiple manifestations and we examined functional and structural changes separately. It is interesting to note that the association was weaker with functional changes (eGFR) measured at a single time-point,

**Table 5 Progression model with top probes associated with gene expression changes**

| Variable[a] | Base model[b] | Base model[b] + kidney promoter probe cg24818418 – Gene EGF (D) | Base model[b] + kidney inactive probe cg21048700 -- Gene COL3A1 (E) | % Explained by variable[c] in base model | % Explained by variable[c] in model D | % Explained by variable[c] in model E |
|---|---|---|---|---|---|---|
| Baseline GFR | 0.03*** | 0.04*** | 0.03*** | 20.67 | 30.78 | 22.03 |
| Diabetes | −0.72* | −1.09*** | −1.17 *** | 5.75 | 8.87 | 10.75 |
| Age | −0.03 | −0.01 | −0.01 | 6.16 | 0.16 | 1.11 |
| CpG probe | NA | 1.24*** | −1.21*** | NA | 17.99 | 13.02 |
| Methylation batch | NA | NA | | NA | 10.71 | 11.26 |
| Bisulfite conversion | NA | 20.28* | 17.70 | NA | 2.90 | 2.46 |
| $R^2$ | 0.51 | 0.75 | 0.70 | | | |
| Adjusted $R^2$ | 0.49 | 0.70 | 0.63 | | | |
| Akaike information criterion | 206.1 | 177.1 | 191.1 | | | |
| P-value | 3.13e−10 | 5.58e−13 | 1.126e−10 | | | |

[a] For each variable, coefficient estimates are shown with the following significance codes: 0 '***'; 0.001 '**'; 0.01 '*'; 0.05 '.'
[b] Model is a weighted linear regression model of adjusted eGFR slope (weight = inverse variance of adjusted eGFR slope). Base model includes variables: baseline eGFR, Diabetes, and Age. Models D and E include base variables with the addition of methylation level at probe location, methylation batch, and bisulfite conversion efficiency
[c] Proportion of variance explained by the variable based on conditional sum of squares calculated in Type II ANOVA analysis

however, the association with structural and longitudinal changes were robust and reproducible. While functional and structural changes strongly correlate, our recent analysis has identified significant differences between functional and structural changes[50]. We found that similar to methylation differences, gene expression changes correlate more strongly with structural changes than with kidney function[50]. Given that eGFR is strongly determined by systemic blood pressure and volume status it does seem to make sense that gene expression and methylation changes have stronger correlation with kidney fibrosis. Renal tubule cells and fibrosis play key roles in DKD development and therefore it is plausible that the association observed in the cell type of interest is more proximal to disease development and stronger.

Our studies indicate not only the robustness of the association but also that methylation changes are likely to be functionally important as they are located on kidney specific regulatory regions and correlate with gene expression changes. Methylation changes are postulated to affect gene expression through interference of transcription factor binding at gene regulatory regions. This relationship is most evident at proximal regulatory regions such as promoters, however, methylation changes at distant regulatory regions (for example enhancers) may similarly affect transcription of nearby genes[41]. In these instances, functional linkage would require experimental confirmation. Regardless, several identified methylation-gene expression pairs appear to be highly interesting. For example, in our samples, decreased methylation at cg20597486 is associated with increased percent interstitial fibrosis on kidney histology and increased expression of *IFI16*, Gamma-interferon-inducible protein. *IFI16* is an important transcriptional regulator and may modulate NF-κB activation[51], which has been shown to play a role in kidney injury in mouse models[52]. Gene ontology analysis shows enrichment of multiple pathways proposed to play important roles in disease development, including development, signaling adhesion and immune system processes. Epigenetic differences in transcriptional regulators could have significant impact on disease development or prenatal or adult programming. However, further testing will be needed to establish causality, for example in a mouse model.

Another key finding of the current work is the identification of methylation changes that can improve kidney function decline models. It is important to note that while the expression of more than a thousand genes correlated with kidney fibrosis at baseline in our study, gene expression levels failed to improve kidney function decline models. Cytosine methylation changes on the other hand were able to improve upon already existing models for kidney function decline,

even though the clinical models show high and reproducible C-statistics.

We would also like to point out an interesting convergence of epigenetic and gene expression changes. Prior functional studies have identified a correlation between kidney *EGF* transcript and fibrosis and urinary *EGF* levels and kidney function decline[53,54]. In our study we replicated the correlation of *EGF* and kidney fibrosis; however, only the methylation status of *EGF*, not the gene expression level, showed significant correlation (p-value = 1.92e−07) with future kidney function decline even after adjustment for baseline GFR, age, and diabetes status. As cytosine methylation levels are relatively stable it makes them potentially ideal disease biomarkers. Gene expression changes follow a minute to minute regulation by environmental factors therefore they might show increased variability and less reproducibility as predictors of progression. The increased stability of DNA over RNA makes methylation levels a more attractive biomarker. Such a finding could have important clinical significance for the development of biomarkers and therapeutics for DKD and our understanding of the pathogenesis of DKD.

While we used microdissected human kidney tubule samples and controlled for as many known and hypothesized sources of variation (i.e., lymphocytic infiltrate) as we could, it is possible that additional unaccounted variation, such as changing cell proportions associated with kidney fibrosis, influenced our results. Despite all our efforts we cannot exclude the contribution of cell heterogeneity to the observed methylation changes. Future studies should aim to control for this by utilizing singe-cell technology that currently exists for RNA sequencing but is not as mature for whole genome epigenome analysis. Additionally, while we aimed to narrow our results by focusing on probes located in active kidney gene regulatory regions, it is possible that these regions are different in the diseased state or in rare cell types such as fibroblasts. For example, we would like to highlight that a top probe cg21048700, that improved the CKD progression model and was associated with changes in *COL3A1* (type III collagen) expression, was not located on regulatory region when whole kidney tubule samples analyzed, however, may still be integral to disease development as this probe is localized on a regulatory region in fibroblasts.

Another potential pitfall of our study is that our longitudinal eGFR samples are from subjects who underwent full/partial nephrectomies that could potentially affect their rate of kidney function decline. However, since all subjects underwent full/partial nephrectomy and we were analyzing relative rate of decline, this likely did not affect our results. In addition, most patients underwent nephrectomy for the indication of renal cell carcinoma, which typically is cured through resection without the

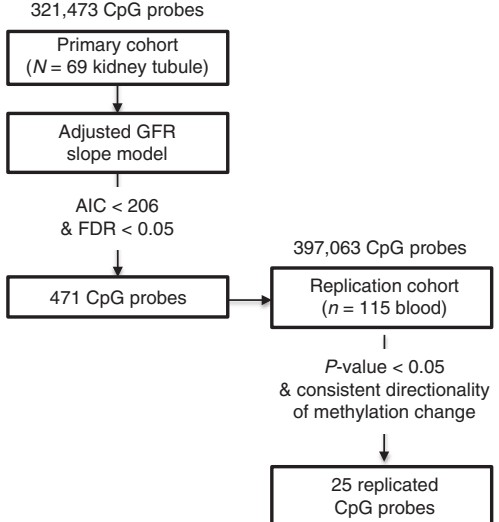

**Fig. 5** Replication of methylation probes associated with kidney function decline 471 methylation probes significantly improved the baseline kidney function decline weighted linear regression model beyond baseline variables (Age, baseline GFR, and Diabetes) (AIC < 206 and FDR < 0.05). Of these 471 probes, 432 probes were assessed in the replication cohort in peripheral blood. 39 probes were excluded due to poor data quality. 52 methylation probes were associated with Adjusted GFR slope when adjusted for Age, Sex, reaching end stage renal disease (ESRD), duration of diabetes, mean blood pressure, hemoglobin A1C, as well as cell type heterogeneity using linear regression (p-value < 0.05). Of these 52 probes, 25 probes maintained directional consistency across the two data sets

need for chemotherapy that can affect kidney function. To support the fact that nephrectomy was not a major determinant of function decline we also show that the rate of progression in our kidney cohort was similar to the one observed for the blood methylation samples and was associated with baseline eGFR and level of interstitial fibrosis on histology, further confirming that GFR decline was mostly determined by the presence of CKD than by the nephrectomy. In addition, changes observed in the kidney cohort were confirmed in the Pima cohort that is an extremely well phenotyped diabetic cohort, without cancer or nephrectomy.

Lastly, because subjects were not enrolled in a prospective study, we cannot account for follow-up data that may have been obtained during acute illness and therefore may reflect acute kidney injury on top of chronic decline. We accounted for these factors by excluding patients without at least three months follow-up and outliers, as well as using an adjusted eGFR slope (using best linear unbiased predictor (BLUP) modeling) in a regression model weighted by the inverse variance of the eGFR slope. Therefore, subjects with a larger degree of variability to their follow-up data were weighted less in our analysis. While some research studies utilize measured GFR, estimated GFR has been shown in clinical studies to be well correlated with outcomes including mortality and end stage renal disease[55]. Finally, there may be other clinical sources of variation (such as medication exposure) not included in our data that could influence our results. Future prospective studies should address these areas.

Despite these limitations we would like to emphasize the robustness of our results as we have validated our findings in an independent kidney cohort, as well as to a lesser degree across tissue samples. Multi tissue transcription factors could explain the replication of differentially methylated loci in multiple cell types. On the other hand, our difficulties in replicating results in surrogate cell types after removal of underlying genetic variability

indicate that sequence variations could also drive cell type independent methylation changes which is in keeping with observations from other diseases[56–59]. Future studies shall aim to dissect the direct contribution of environmental factors and sequence variations in cytosine methylation in the kidney. Despite these limitations, here we identified pathways and replicated previously observed pathways associated with kidney disease pathogenesis and progression. Given the correlation with the gene expression pattern and the association with future functional decline it is likely that some of these methylation changes are also functionally important for DKD development.

Overall we show genome wide significant and replicated cytosine methylation differences in DKD and use machine learning methods to identify loci that can improve current models for kidney function decline. These loci should be studied in future prospective cohort studies as candidate biomarkers or pathways for intervention.

## Methods

**Participants**. The primary cohort included 91 human kidney samples obtained from surgical full/partial nephrectomies. Kidney samples were ~0.5 cm in diameter and were surrounded by at least 2 cm of normal tissue margins. The study used cross-sectional design. The samples were collected from Albert Einstein College of Medicine Montefiore Medical Center between the years of 2007–2011. Samples were de-identified and the corresponding clinical information available at the time of nephrectomy was collected by an honest broker. Hypertension and diabetes diagnosis were based on chart review diagnosis codes. Blood pressure values were obtained at the time of tissue procurement. In this cohort, 41 subjects carried a diagnosis of DM and 65 carried a diagnosis of HTN and there were 22 patients with diabetic CKD. Estimated GFR was determined using the Chronic Kidney Disease Epidemiology Collaboration (CKD-EPI) estimating equation. In this primary cohort, all subjects with GFR < 60 had diabetes and evidence of DKD on histologic analysis (n = 22). Subjects with GFR > 60 included those with and without diabetes and with and without hypertension. Clinical data were obtained at the time of the sample collection and for a subset of samples estimated glomerular filtration rate (eGFR) measures were available from clinical records before and after the nephrectomy (n = 69). Only subjects with longitudinal eGFR measurements for at least three months post nephrectomy were included for the analysis. The mean timespan of follow-up was 2.4 years (SD = 1.5 years).

Institutional Review Boards at the Albert Einstein College of Medicine Montefiore Medical Center (IRB 2002–202) and the University of Pennsylvania (IRB 815796) reviewed this study. This project utilized de-identified kidney biospecimens. Therefore, this project does not meet the definition of human subject research and IRB review was not required. It was completed in compliance with all relevant ethical regulations.

The replication cohort included 85 microdissected human kidney tubule samples obtained from surgical full/partial nephrectomies. The cause of CKD in this cohort was mixed diabetic and hypertensive kidney disease. There were 10 patients with diabetic CKD and 38 patients with non-diabetic CKD (where CKD corresponds with eGFR < 60 ml/min per 1.73 m$^2$). 11 patients had diabetes and 25 had hypertension in the absence of CKD. The replication data set included 23 samples from our primary data set in order to assess degree of batch effect.

The second replication cohort was comprised of 115 peripheral blood samples obtained from American Indians with DKD enrolled in a longitudinal study described by Qiu et al.[6] Blood samples for DNA isolation and cytosine methylation analysis were collected at the baseline examination in patients with diabetes and chronic kidney disease (albumin to creatinine ratio ≥ 300 mg/g or eGFR < 60 ml/min per 1.73 m$^2$). The mean timespan of follow-up was 5.6 years (SD = 3.5 years). 45 cases reached the endpoint of end stage renal disease.

**Procedures**. A portion of the harvested kidney was formalin fixed and paraffin embedded and stored for Period Acid Schiff and Hematoxylin and Eosin staining. The histopathology was evaluated by a blinded pathologist and described using a descriptor method scoring 19 independent parameters, including the degree of tubulointerstitial fibrosis. Histopathological lesions were used to determine the cause of CKD in our study. Seven samples in the primary and eight samples in the replication cohort had missing fibrosis score.

The rest of the kidney was freshly immersed into RNALater and stored at -80C. Samples were manually microdissected into glomerular and tubular compartments under a stereomicroscope using established methods. DNA and RNA was then isolated from tubule tissue using the Qiagen RNAeasy and DNAeasy or AllPrep DNA/RNA mini kits following manufacturer's instructions. Transcript level changes were determined using the Affymetrix U133 RNA microarray. DNA was bisulfate converted using the EZ DNA methylation kit (Zymo research) and assayed using the Illumina Infinium HumanMethylation450K BeadChip according to the manufacturer's instructions. The bisulfite conversion efficiency was

calculated using the bisulfite conversion control probes, based on Illumina guidelines. Ten CpG sites designated by Illumina as control sites (6 CpGs targeted by type I probes and 4 CpGs targeted by type II probes), where we expected each CpG to be 100% methylated, were used to control for non-complete bisulfite conversion. The bisulfite conversion efficiency used in the primary analysis was the median methylation estimate from the ten control sites. The bisulfite conversion term was calculated by taking the median value of the probes that Illumina provides to estimate bisulfite conversion efficiency[61]. Each cohort (primary and replication cohorts) was run as a single batch at different facilities. Therefore, they were initially analyzed separately and then combined to avoid capturing facility-batch artifact[58]. Of note, all analysis were additionally controlled for batch effect as defined by Illumina slide position (Sentrix ID).

DNA methylation values were extracted using the minfi package[62]. Our methylation analysis pipeline included several quality control measures. We used the Gap Hunter package[63] to remove CpG probes that are in the proximity of regions with genetic variations (gap probes). We also removed probes located on the sex chromosome and known to cross-hybridize to other locations. We also excluded probes with poor detection $p$ value ($p > 0.01$) and control probes. Finally, we were left with 321,473 CpG probes in the primary cohort. The same preprocessing was performed for the kidney replication cohort and resulted in 355,141 CpG probes in the kidney replication cohort. The blood replication cohort was processed in the same pipeline with the exception of application of the Gap Hunter program−instead, probes located within 1 bp of common single nucleotide polymorphism (SNP) loci (with minor allele frequency > 0.01 according to dbSNP137) were removed. Finally, there were 401,438 CpG probes in the blood replication cohort. Principal component analysis was performed on each cohort and outlier samples were removed (one sample from the primary cohort, two samples from the kidney replication cohort and two samples from the blood replication cohort). In order to account for probe bias inherent to the 450 K array, we then normalized methylation data utilizing beta-mixture quantile normalization method through the r package, RnBeads, resulting in normalized β values for each CpG probe[64]. β values represent the methylation level and range from 0 to 1 for unmethylated to methylated loci, respectively. Due to the fundamental heteroscedacity to beta values, these values were log transformed to M values (M = $\log_2(\beta/(1 - \beta))$) for use in all linear regression models[65].

Human kidney histone ChIP-Seq data including H3K4me1, H3K4me3, H3K27ac, H3K36me3, H3K9ac, and H3K9me3 (GSM621634, GSM621648, GSM621651, GSM670025, GSM772811, GSM1112806) was used for functional genomics annotation. We used chromHMM, software that integrates epigenome marks based on a multivariate Hidden Markov Modeling. The resulting model can be used to systematically annotate the genome such as promoter, enhancer, transcribed, and inactive (quiescent, heterochromatin, bivalent, and repressed) states[66].

Subject specific unadjusted eGFR slopes were determined by linear regression across all available eGFR measures. Only subjects with a minimum of 3 eGFR estimations and longitudinal eGFR measurements for at least three months post-nephrectomy were included for the analysis. Three months was chosen as a minimum as a way to minimize the acute changes peri-nephrectomy. Subjects with unadjusted eGFR slope < −40 or >40 ml/min per 1.73 m² per year were excluded. In order to account for random variation as well as non-normal distribution of the data, subject specific eGFR slope was adjusted using a form of mixed effect model, best linear unbiased predictor (BLUP)[67], using the R package lme4. Subject specific adjusted eGFR slope and variance were used in weighted regression analysis to examine the association with methylation levels and other predictors of progression. The regression was weighted by the inverse variance of the slope, such that subjects with increase variability to their eGFR slope were weighted less in the analysis.

**Statistical analysis.** Linear regression models were used to determine the association between cytosine methylation level (at each CpG probe), kidney function (eGFR) and structural damage (measured by interstitial fibrosis). The model was adjusted for age, sex, race, diabetes, hypertension, batch effect, bisulfite conversion efficiency, and degree of lymphocytic infiltrate on histology. False Discovery Rate (FDR) was used to account for multiple comparisons and determine epigenome wide statistical significance[68]. Based on power estimation published by Tsai et al.[69], we estimated that with approximately 1:3 case to control ratio, that we had >80% power to detect methylation difference 20–25 and 100% power to detect mean methylation difference 30% in our primary cohort.

To understand the association between clinical and histological variables and kidney function decline first the association between each variable and adjusted eGFR slope was examined. Variables that showed a significant ($p < 0.05$) association with adjusted eGFR slope were later included for the machine learning regression analysis method: least absolute shrinkage and selection operator (LASSO)[44]. To implement LASSO we utilized the R program glmnet using the gaussian response family. LASSO selected variables were used to create a CKD progression base model. Adequacy of model fit was determined by Akaike Information Criterion (AIC). CpG probe methylation was individually added to the base model using weighted linear regression (weight = inverse variance of adjusted eGFR slope) and additionally controlled for batch effect and bisulfite conversion efficiency. False discovery rate (FDR) procedure was used to account for multiple comparisons[70]. Methylation Probes that improved model fit (AIC lower than base

model) and reached epigenome wide statistical significance (FDR < 0.05) by weighted linear regression analysis were further analyzed.

Weighted linear regression was also used to analyze the association between cytosine methylation and adjusted eGFR slope (weight = inverse variance of adjusted eGFR slope) in our blood replication cohort. Covariates in this regression model included age, sex, duration of diabetes, mean blood pressure, hemoglobin A1c, batch effect, bisulfite conversion efficiency as well as distribution of cell types[60].

**Reporting summary**. Further information on research design is available in the Nature Research Reporting Summary linked to this article.

## Data availability
The methylation data is available at GEO: GSE50874. The gene expression data is available at ArrayExpress: E-MTAB-5929, E-MTAB-2502. The rest of the data are available from the corresponding author upon reasonable request.

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

## Acknowledgements

This work was done with the support of NIH NIDDK RO1 DK087635, DP3 DK108220 to K.S., F32 DK112635 to C.G., American Diabetes Association to K.S. and R.H. The funders had no role in study design, data collection and analysis, decision to publish, or preparation of the manuscript.

## Author contributions

C.G. and C.Q. developed the methylation analysis pipeline and performed data analysis. S.Y.H., A.S.D.P., and M.C.I. microdissected kidney samples and produced kidney cytosine methylation and gene expression data. R.H. collected American Indian cohort with blood methylation data. M.P. and J.P. scored kidney histology. I.M. and A.V. coordinated clinical data. C.G., Y.K., and J.P. performed kidney histone annotation. C.G., C.Q., X.S., J.H., Y.C., and H.L. performed statistical analysis of methylation data and kidney function progression data. Y.G. assisted with PCR replication of top methylation probe. K.S. provided oversight analysis and coordination of all aspects listed above. C.G. and K.S. wrote the manuscript. All authors reviewed the manuscript and provided comments.

**Additional information**

**Competing interests:** The authors declare no competing interests.

