## [Peer Review File · Nature Communications]

Reviewers' comments:

Reviewer #1 - expert in EWAS (Remarks to the Author):

Kidney functional decline is a frequent disease sequela of diabetes and leads to significant morbidity and mortality. However, after diagnosis due to poor metabolic indices, robust long-term glycaemic control does not appreciably reduce the risk of renal disease.

Gluck *et al.* have proposed that the initial metabolic abnormalities impact epigenetically to maintain this pathogenic trajectory. In this study they have analysed the DNA methylation and gene expression of micro-dissected human kidney tubule epithelial cells derived from 91 subjects with Illumina 450k DNA methylation arrays and Affymetrix U133 RNA microarrays, respectively. This study set included 45% with diabetes, 71% with hypertension, and with varying degrees of kidney disease. In this set they initially identified 518 cytosines with Bonferroni significant DNA methylation changes associated with interstitial fibrosis. These results were replicated in a further 85 samples with chronic kidney disease in 459 cytosines with nominal significance and directional consistency. When combined, 279 cytosines were identified to be significant with respect to interstitial fibrosis and these loci were enriched within kidney enhancer regions from ChromHMM Segmentation data.

In a model of renal functional decline (adjusted eGFR slope), 1,131 cytosines captured kidney deterioration, which they found was not possible with either histological or gene expression data. 73 of these cytosines were replicated in peripheral blood derived DNA from an American Indian cohort, being nominally significant with consistent directional change. These CKD progression model fit cytosines were also compared with a recent study in blood from diabetes and 2 cytosines were also differentially methylated.

This study overall has significant strengths, particularly due to its direct exploration of the disease-relevant kidney cells and not a surrogate tissue, as well as additional replication/validation analyses. However, there are several points below that I would like the authors to address to further substantiate their claims.

Major

- 1) The numbers analysed are small by array standards, although, this is obviously restricted by the ability to access kidney tissue samples. Yet, this will still impact on the power of this genome-wide study. The use of a multi-ethnic sample group will also significantly increase the potential of genetic heterogeneity to confound or inflate results. The authors have incorporated a race category into their linear regression model; however, common genetic background is difficult to correct for using only broad racial/ethnic groups. Genetic and cell-type heterogeneity can drive significant variation and, for example, minimal DMPs are identified when genetics and cell-type is controlled for in isolated cell-type monozygotic twin studies (1). Did the authors attempt any further exploration of potential genetic effects, such as "gap hunting" (2), incorporating any genetic

information on these individuals, or else? The authors indicate that CpG probes “near” common SNPs were removed – can they state this more precisely: was it 10 bp or else? The Manhattan plot (Fig. 1a) does appear inflated for low p-values --- see Figure 1e

in Lunnon *et al.* (2014) (3) for an EWAS in Cortex tissue for comparison. Did the authors assess potential inflation by a qqplot?

- 2) Can the authors comment further on cell type heterogeneity issues with respect to the analysis within the kidney samples – inflammatory cell infiltration *etc.*
- 3) Replication is supportive, however, for those results identified in whole blood (American Indian and Chen *et al.*) --- what is the pathophysiological mechanism to explain the commonality between tissues beyond genetic confounding? Was the number of overlapping probes above expected randomly?
- 4) The authors propose a hypothesis whereby the epigenetic state of the kidney cells acts as a metabolic memory primed by the early adverse metabolic effects. Another further explanation is the impact of ageing on these initial metabolically---driven epigenomic abnormalities, as the epigenome changes with age, both stochastically (4), but also directionally within certain functional loci (5, 6). The authors have excluded ageing change, but this may further perturb these abnormalities. As well, other age---related epigenetic changes related to kidney disease may exist, perhaps in conjugation with genetic disease susceptibilities. This could be attempted to be explored in these data as there is an S.D. of 11.5 in age. Are there any significant age---related DNA methylation changes in epigenetically defined kidney regulatory regions? Or Diabetic kidney disease GWAS loci?
- 5) Could the authors provide more details as to how the enrichment within enhancers was calculated? Low levels of variation in promoters is expected as these CpG---dense regions are predominately non---variant (7) – was this accounted for?
- 6) 279 probes were identified as enriched for kidney enhancer regions. Whilst enhancers are significantly more tissue---specific than promoters, was this result definitively specific to kidney tissue? --- or are these probes also enriched for tissue common enhancer regions --- as can be evaluated in the other available Encode ChromHMM Segmentation data, for example?
- 7) Can the authors include more details as to the expression analysis and how multiple testing was accounted for? If a random permutation of these data is performed, how many significant associated expression results are identified?

Minor

- 1) Introduction pg 4: Sentence “since the epigenome is under... “ – requires re--- writing
- 2) Introduction pg 5: missing “to” --- “...failed to pass...”
- 3) Results pg 6/7: “bisuphite conversion efficient” not “methylation conversion efficiency” as it is unmethylated cytosines that are converted

- 4) Results pg 7: "... histone tail modifications ..."
- 5) Methods pg 16: "bisulphite" not "bisulphate"

1. Paul DS, Teschendorff AE, Dang MA, Lowe R, Hawa MI, Ecker S, et al. Increased DNA methylation variability in type 1 diabetes across three immune effector cell types. *Nat Commun.* 2016;7:13555.
2. Andrews SV, Ladd-Acosta C, Feinberg AP, Hansen KD, Fallin MD. "Gap hunting" to characterize clustered probe signals in Illumina methylation array data. *Epigenetics & chromatin.* 2016;9:56.
3. Lunnon K, Smith R, Hannon E, De Jager PL, Srivastava G, Volta M, et al. Methylomic profiling implicates cortical deregulation of ANK1 in Alzheimer's disease. *Nat Neurosci.* 2014;17(9):1164-70.
4. Issa JP. Aging and epigenetic drift: a vicious cycle. *J Clin Invest.* 2014;124(1):24-9.
5. Teschendorff AE, Menon U, Gentry-Maharaj A, Ramus SJ, Weisenberger DJ, Shen H, et al. Age-dependent DNA methylation of genes that are suppressed in stem cells is a hallmark of cancer. *Genome Res.* 2010;20(4):440-6.
6. Rakyan VK, Down TA, Maslau S, Andrew T, Yang TP, Beyan H, et al. Human aging-associated DNA hypermethylation occurs preferentially at bivalent chromatin domains. *Genome Res.* 2010;20(4):434-9.
7. Ziller MJ, Gu H, Muller F, Donaghey J, Tsai LT, Kohlbacher O, et al. Charting a dynamic DNA methylation landscape of the human genome. *Nature.* 2013;500(7463):477-81.

Reviewer #2 - expert in DNA methylation and diabetes (Remarks to the Author):

Clearly a study that embarks on distinguishing molecular signatures from human kidney tubule tissues is to be applauded. Rather than focus on single genes the authors discuss in the introduction the multifactorial contribution of environment and genes working in pathological states. This paper is the beginnings of a significant study that contributes to the establishment of molecular diagnostic criteria for the classification of diabetic kidney disease (DKD). Kidney function decline constitute a major proportion of intractable diabetic complications and the classification and diagnosis of DKD remains a clinical challenge. Gluck et al use array-based technologies to derive DNA methylation and gene expression using surgically resected kidney tubule tissue from human subjects with and without diabetes. Replication cohorts include kidney tubule tissues as well as blood derived methylation assessments. The major finding of this paper is the identification of specific DNA methylation signatures that could be used to predict kidney function decline. In striking contrast, differences in gene expression did not match with DKD subtypes and were likely driven by unknown factors that are not explored in any detail. DNA methylation is unsurprisingly a stable biological mark when compared to mRNA derived assays. The implications of these findings are unclear. The paper begins to add differential DNA methylation signatures as a potential molecular classification for DKD.

From a technical point of view the study was performed using 450K array and is limited to predetermined probe sets as discussed by the authors. More significant here and a point of confusion for the Nature Commun readership is the approach used is not “genome wide” methylation assessment. There is also confusion and lack in clarity in the analytical approaches used which are of concern. And while my enthusiasm is high for the article there is a paucity in mechanistic discovery. The methylomes assessments are observational and descriptive. And while there is appreciation that gene contribution is likely to be multifactorial few genes standout as strong methyl-biomarkers of the study.

The paper could further be strengthened by considering the following:

1. While it is helpful to discuss potential mechanisms implicated in DKD that may translate into differential DNA methylation changes, what are the factors that influence DNA methylation in kidney disease? This is central to this study. Could DNA methylation be regulating pathways central to kidney disease and how could this information be used to improve predictive utility?
2. The authors are encouraged to examine whether differential DNA methylation patterns could also be used as a biomarker for the predictive classification of DKD (i.e. tubules) and from blood samples. Are the mechanisms that drive DNA methylation in kidney and blood generalizable? The current study does not address mechanism which would be a significant advance in the field.

However, this is not the only major criticism of the study and the comments below address some of the issues.

Abstract:

Authors describe “genome wide methylation” in methods, this examination is by array, which is not described and the Illumina 450 array is not considered “genome wide” which is usually reserved for CpG methylation sequencing or whole genome bisulfite seq WGBS based methods. The article requires clarity on the actual DNA methylation method used, as readers will be confused throughout the manuscript.

Methods abstract, the authors describe gene expression and histone maps for functional integration, this requires a cleaner description, as it does refer to ENCODE datasets assessed for epigenetic changes, not necessarily derived from the study. This description needs to read more accurate as it implies further epigenetic assessments and analyses were derived from the tissues.

Introduction:

Page 4 in the 3rd paragraph, the authors discuss intrauterine exposure and kidney disease development. The link here with development is unconvincing. Are the authors implying intrauterine DNA methylation is associated with DKD? The articles cited do not conclusively illustrate this point.

Methods/Design:

How were the independent cohort selected? were they from different centres? and how many years apart? How is the blood replication cohort relevant when considering these samples are from American Indians? This is not only confusing but also raises other salient questions regarding the generalizability of the DKD methylation marks from the different cohorts.

Is there any genetic data available for the samples used in this study? Is it possible that genetic heterogeneity due to population genetic variation in CpG-SNPs could be driving methylation differences?

A combination of Bonferroni and FDR correction was used for Multiple comparisons throughout the study. For consistency in statistics can the authors use one or the other? The paper does not estimate what statistics were used for correction.

Did the authors perform any power analysis prior to the DNA methylation analysis?

No methyl-validation of probes were attempted and the study requires stronger assessment of methylation difference using an independent assay. The differences in the methylation levels of the CG probes could be a function of the array.

Results:

The main research question for this manuscript is to identify cytosine methylation difference in patients with diabetic kidney disease, how did the authors adjust for diabetes?

Could the authors assess the effect of gender, age and race on the differentially methylated probes?

Odds ratio analysis was used to define functional importance of methylation differences. ChromHMM uses background data to calculate overrepresented in differential data, can the authors give details on the background dataset used?

Gene expression was also performed in (n=58), however the results are not shown. One of the aims of the study was integrated methylation changes with regulatory and gene expression changes, yet there are no results shown. Authors only show gene expression of a few selected genes. If this was not the aim of the study than it remains confusing what hypothesis the authors are assessing.

Page 7, how was linear regression adjustments made for age, sex, race, diabetes, hypertension and batch effect? Description of the methylation conversion efficiency is also warranted showing how this was performed.

Page 7, "to better control for batch effect" this is unusual and it's unclear why this was not performed earlier in the paper.

Page 7, ChromHMM integration performed on datasets derived for the histone modifications in "human kidney" presumably they mean non-DKD tissue? If so, how are the epigenetic comparisons made for diabetic and non-diabetic kidneys?

Page 7, the first description of the array is in the results section of the paper.

Page 8, the paper needs to define the “stringent statistical criteria” used for methylation and gene expression changes. What are they and better descriptions required to justify methylation differences.

Page 8, methylation is inversely correlated for HOPX gene expression. The importance of this finding in the paper is not strong. What available data is there that HOPX is regulated by DNA methylation driving gene expression?

Page 9, “the methylation level of 1,131 probes was significantly associated with renal function decline...” unclear in which cohort the result is observed.

Discussion:

Page 12, “robustness including blood sample from the DCCT study” ... Yet the overlap between data generated in this study and DCCT is limited to 2 CpG sites (page 11). The paper does not illustrate robustness of the association, rather, observation the 450K array identifies methylation differences at probes that might be considered passing statistical testing.

Page 12, it remains unclear how the methylation changes for the genes screened for methylation difference, such as HOPX are functionally important.

Figures/Tables:

Table 1, Diabetes duration not shown. Over how many years and measurements was eGFR assessed?

Table 3, can authors report effect size of methylation change for the probes shown?

Figure 3a and 4a, can authors show confidence intervals on plots?

No description table for gene expression cohort is shown.

Suppl. Information:

The paper should show methylation data from all samples using PCA plots in order for the reader to visualize the variability in methyl-signals.

Reviewer #3 - expert in diabetic nephropathy (Remarks to the Author):

In this retrospective, observational analysis Gluck and coworkers tried to define the wide cytosine methylation differences in microdissected human kidney tubule epithelial cells of patients with diabetes and kidney disease and to evaluate whether and to what extent analysis of cytosine methylation levels can improve the predictive value of current models of renal function decline. The issue is of potential methodological and clinical interest. Unfortunately, the study is flawed by major limitations in patient selection and outcome analyses, the analyses fail to address the primary question of the paper, data are unclear and their presentation is confusing and potentially misleading.

More specifically:

1. The study population is poorly characterized. Main clinical and laboratory characteristics at the time the kidney tissue was sampled should be provided in Table 1, including detailed information about factors that may affect renal disease outcome such as blood pressure and metabolic control, serum lipids, underlying histological diagnosis of kidney disease, concomitant treatment with drugs such as ACE inhibitors, ARBs, HMGCoA inhibitors, blood glucose and blood pressure lowering agents and others that may affect GFR decline over time (the primary outcome of the study). All these parameters should be considered in the univariable and multivariable models used by the Authors.

2. The title and data presentation throughout the text are misleading. Why the Authors focus the attention on diabetic kidney disease when only 41 of the 91 study patients were diabetics?
3. Again, the Authors state that methylation differences with genome wide significance can be detected in kidney tubule samples of patients with diabetic kidney disease. Thus, why data on 50 non-diabetic patients are reported in Table 1? Information about non diabetic patients should be deleted.
4. Independent of the above, the Authors should acknowledge that the study population is small, which may affect the power of the analyses and the robustness of the findings. In this context, a justification for the sample size should be provided to ensure that study findings are not casual.
5. The authors do not take into consideration that type 1 and type 2 diabetes are two different disease entities, with different etiologies, phenotypes, outcomes and treatment. In type 1 diabetes hyperglycemia is the direct consequence of impaired insulin production, in type 2 diabetes hyperglycemia is just one of the several manifestations of the metabolic syndrome, including hypertension, dyslipidemia, obesity, insulin resistance and other. Thus, the Authors should clarify how many of their patients had type 1 or type 2 diabetes and should consider them separately.
6. The definition of diabetic kidney disease is also nebulous. In patients with type 2 diabetes, kidney changes may include typical focal glomerular sclerosis (in a minority of cases), nephroangiosclerosis, ischemic kidney disease, tubule-interstitial disease, concomitant primary glomerular diseases, ageing-related changes and other changes that can be observed in different combinations in different patients. Probably these different hystological patterns reflect different pathogenic mechanisms. Thus it is hard to believe that a single predictive model may predict disease outcome to a similar extent in such a heterogeneous population of patients. This is an issue that should be taken in due consideration by the Authors and should be adequately discussed. In table 2 the authors should report the histological diagnosis of the study patients.
7. More in general no information is provided about the selection criteria for study participation. According to data in Table 1 it is conceivable that a subgroup of “healthy” subjects without diabetes, hypertension or proteinuria was also included. This should be clarified. How can methylation changes in “healthy” subjects predict the outcome of diabetic kidney disease? Incidentally, reasons for nephrectomy should be reported.
8. More detailed information should be provided about outcome analyses. How eGFR slopes were calculated? Did the authors account for the acute changes associated with nephrectomy (sharp

GFR reduction after nephrectomy followed by compensatory hyperfiltration of residual tissue) and may be subsequent decline due to exhaustion of surviving nephrons? Which was the minimum number of GFR estimations and the minimum follow up required for slope calculations?

9. The study design should be described in title and abstract; study setting, participating centers, recruitment period, data monitoring should be described in detail in the text. A justification of the sample size should be provided. Was an informed consent required for the use of kidney tissue for the purposes of the study? Was the protocol approved by an ethical committee?

10. Although I recognize the difficulty in finding biopsy samples, the validation step for the methylation changes that predict kidney function decline in peripheral blood mononuclear cells (PBMCs), instead of microdissected tubules, is not fully convincing. Indeed, one of the first assumptions is that methylation is cell type-specific. What is more, one of the top probes found to correlate with kidney function decline is not confirmed by results published by Chen et al. on whole blood from patients treated with conventional glycemic therapy and diabetic complications. To use PBMCs in the validation step, the authors should confirm that microdissected tubule methylomes match (fit) with the PBMCs ones in the primary and replication cohorts.

11. Locus specific validation of kidney cytosine methylation and gene expression changes for the top replicated probes associated with interstitial fibrosis and kidney function decline are lacking and should be provided through bisulfite sequencing and real time qPCR experiments, respectively.

12. The authors identified 7 probes associated with interstitial fibrosis, but described only the one correlated with HOPX transcript level. It is not clear why they omitted the other descriptions. The authors should discuss the possible involvement of all the differentially methylated and expressed genes (listed in Table 3) in the progression of interstitial fibrosis in DKD. Moreover, in vitro experiments in proximal tubule cells exposed to high glucose or silencing/overexpressing HOPX are needed to elucidate the involvement of HOPX in interstitial fibrosis.

13. Similarly to point 2, the authors should describe other possible candidates (besides EGF) as predictors/biomarkers for renal function decline, focusing on those that are known to play a role in kidney patho-physiology (for example collagen, as shown in Supplementary Table 6).

14. The abstract results are misleading. It seems like methylation changes in the kidney promoter regions next to EGF were validated in blood samples of the independent cohort, while they were not.

Minor points:

a) Several inaccuracies occur throughout the text:

- Is the replication cohort left with 416354 CpG probes (as stated in the Procedures section) or 406354 (as stated in Figure 2)
- In the last paragraph on page 13, the authors first assert that subjects underwent full/partial nephrectomy, but then that all subjects underwent partial nephrectomy. Please correct accordingly.
- In the abstract, the authors should substitute diabetes with diabetic kidney disease (page 2, line 9)
- On page 10, line 24, the authors reported 73 CpG probes, while in Figure 5 and its legend, 76 are reported. Please correct accordingly.

b) Results and Procedures should be divided into paragraphs and titled in order to facilitate reading of the paper.

c) The authors should specify in greater detail which Affymetrix microarray they used.

Reviewer #1 – expert in EWAS (Remarks to the Author):

Kidney functional decline is a frequent disease sequela of diabetes and leads to significant morbidity and mortality. However, after diagnosis due to poor metabolic indices, robust long-term glycaemic control does not appreciably reduce the risk of renal disease.

Gluck et al. have proposed that the initial metabolic abnormalities impact epigenetically to maintain this pathogenic trajectory. In this study they have analysed the DNA methylation and gene expression of micro-dissected human kidney tubule epithelial cells derived from 91 subjects with Illumina 450k DNA methylation arrays and Affymetrix U133 RNA microarrays, respectively. This study set included 45% with diabetes, 71% with hypertension, and with varying degrees of kidney disease. In this set they initially identified 518 cytosines with Bonferroni significant DNA methylation changes associated with interstitial fibrosis. These results were replicated in a further 85 samples with chronic kidney disease in 459 cytosines with nominal significance and directional consistency. When combined, 279 cytosines were identified to be significant with respect to interstitial fibrosis and these loci were enriched within kidney enhancer regions from ChromHMM Segmentation data.

In a model of renal functional decline (adjusted eGFR slope), 1,131 cytosines captured kidney deterioration, which they found was not possible with either histological or gene expression data. 73 of these cytosines were replicated in peripheral blood derived DNA from an American Indian cohort, being nominally significant with consistent directional change. These CKD progression model fit cytosines were also compared with a recent study in blood from diabetes and 2 cytosines were also differentially methylated.

This study overall has significant strengths, particularly due to its direct exploration of the disease-relevant kidney cells and not a surrogate tissue, as well as additional replication/validation analyses. However, there are several points below that I would like the authors to address to further substantiate their claims.

We would like to thank the reviewer for his/her positive comments.

Major

1) The numbers analysed are small by array standards, although, this is obviously restricted by the ability to access kidney tissue samples. Yet, this will still impact on the power of this genome-wide study. The use of a multiethnic sample group will also significantly increase the potential of genetic heterogeneity to confound or inflate results. The authors have incorporated a race category into their linear regression model; however, common genetic background is difficult to correct for using only broad racial/ethnic groups. Genetic and cell-type heterogeneity can drive significant variation and, for example, minimal DMPs are identified when genetics and cell-type is controlled for in isolated cell-type monozygotic twin studies (1). Did the authors attempt any further exploration of potential genetic effects, such as “gap hunting” (2), incorporating any genetic information on these individuals, or else? The authors indicate that CpG probes “near” common SNPs were removed – can they state this more precisely: was it 10 bp or else? The Manhattan plot (Fig. 1a) does appear inflated for low p-values - see Figure

1e in Lunnon et al. (2014) (3) for an EWAS in Cortex tissue for comparison. Did the authors assess potential inflation by a qqplot?

Thank you for pointing this out. As the reviewer notes, it is exceedingly difficult to obtain human kidney biopsy samples for epigenetic studies. Indeed, we are not aware of any other cohorts or publications that used the direct tissue of interest for their analyses. We have tried our best to have the largest possible sample size. This sample size is comparable to other tissue based EWAS studies, but indeed lower than surrogate cell-type (blood) EWAS publications. Regardless of these issues we believe that tissue based analyses are essential and complementary to large surrogate cell type analysis. Our initial goal was to identify methylation markers for disease (fibrosis) regardless of race or cell types, however the reviewer is making an excellent point about inflation. The original analysis qqplot indicates significant inflation ($\lambda = 2.73$).

To assess the cause of the inflation, first we used the gap hunter program (as suggested) to remove probes that might be identified as differentially methylated in fibrosis due to differences in genetic background. In our dataset the gap hunter program identified 110,599 “gap signal” probes for removal. Removing these probes, however, this did not improve the qqplot for the association of methylation changes with degree of kidney interstitial fibrosis, indicating that genetic background was not the main determinant of the inflation.

As also suggested by the reviewer, we next hypothesized that cell type heterogeneity could contribute to this inflation, despite our efforts to use microdissected human kidney tubule samples. Specifically, we noted that in lymphocytic infiltrate could be a major contributor of cell heterogeneity. Therefore, we included degree of lymphocytic infiltrate in our regression and found it had dramatically reduced the inflation ($\lambda = 1.70$). Therefore, we re-analyzed our entire dataset by including lymphocytic infiltrate into our model.

Our final model included the following covariates: age, sex, race, hypertension, diabetes, bisulfite conversion efficiency, batch, and lymphocytic infiltrate. Using this improved model, we correlated the degree of interstitial fibrosis with methylation level (M value) by analyzing the methylation of remaining 321,473 probes, after “gap probe” removal. We found that in our primary data set, the methylation level of 203 probes were still significantly associated with interstitial fibrosis ($FDR < 0.05$).

We applied this improved model to our replication kidney cohort and well as our combined cohort. When examining only the 203 significant probes from our primary cohort analysis, we found that 65 probes were also associated with interstitial fibrosis in our secondary cohort (p -value < 0.05 with consistent direction of methylation change) as well as our combined cohort ($FDR < 0.05$). We then continued the downstream analysis of these 65 replicated probes.

Next, the same approach was used to identify probes those methylation correlates with kidney function decline. We again removed probes that might have been related to genetic background, using the Gap Hunter method. We found that this improved our qqplot (λ decreased from 2.02 (left) to 1.85 (right)).

We did not, however, include lymphocytic infiltrate as a covariate since the goal of our analysis was to identify biomarkers of progression. While both interstitial fibrosis and lymphocytic infiltrate were associated with progression (adjusted eGFR slope) by univariate analysis, when we applied the machine learning method (LASSO) to choose variables associated with progression, these variables were not selected. We had originally identified 1,131 methylation probes that improved our progression model (AIC < 206 and FDR < 0.05). However, a significant number of these probes (406) were removed by the gap hunter program due to potential genetic variance. When we re-ran the analysis without these “gap probes”, we found 471 probes that improved our model (AIC < 206 and FDR < 0.05), including the probe associated with epidermal growth factor (EGF) expression changes.

2) Can the authors comment further on cell type heterogeneity issues with respect to the analysis within the kidney samples – inflammatory cell infiltration etc.

We appreciate this comment. We hypothesized that lymphocytes potentially “contaminated” the microdissected tubule cell data since degree of lymphocytic infiltrate correlates with percent interstitial fibrosis on histology (cor = 0.723, p-value = 1.092 E - 14). We therefore included lymphocytic infiltrate in our regression analysis. Please see our response above as we re-ran the analysis to include cell type heterogeneity in our model. We narrowed our top probes by those located in active kidney gene regulatory regions. We discuss the issue of cell type heterogeneity and its role in the dataset.

3) Replication is supportive, however, for those results identified in whole blood (American Indian and Chen et al.) - what is the pathophysiological mechanism to explain the commonality between tissues beyond genetic confounding? Was the number of overlapping probes above expected randomly?

Thank you for pointing this out. There is a large body of literature on using surrogate cell types to define epigenetic changes that are observed in disease relevant cell types. For example, in the Alzheimer literature some methylation changes observed in blood samples correlate with brain specific changes^{1,2}. Similar observations have been published in the obesity literature³. It seems that changes observed in surrogate cell types are less pronounced, similar to our observations. One possible hypothesis could be that the changes are caused by specific transcription factors that are independent of

cell types.

Was the number of overlapping probes above expected randomly?

Our original analysis identified 1131 probes that improved our CKD progression model. Out of these 1131 probes, 135 probes were similarly associated with CKD progression in the replication cohort analyzing blood samples. If we randomly selected 1131 probes from the entire array ten thousand times, the likelihood of replicating 135 probes in the blood replication cohort was low (permutation p-value = 0.011).

4) The authors propose a hypothesis whereby the epigenetic state of the kidney cells acts as a metabolic memory primed by the early adverse metabolic effects. Another further explanation is the impact of ageing on these initial metabolically-driven epigenomic abnormalities, as the epigenome changes with age, both stochastically (4), but also directionally within certain functional loci (5, 6). The authors have excluded ageing change, but this may further perturb these abnormalities. As well, other age-related epigenetic changes related to kidney disease may exist, perhaps in conjugation with genetic disease susceptibilities. This could be attempted to be explored in these data as there is an S.D. of 11.5 in age. Are there any significant age related DNA methylation changes in epigenetically defined kidney regulatory regions? Or Diabetic kidney disease GWAS loci?

Thank you for this important question. To understand age associated methylation changes in kidney samples we performed analysis on healthy controls samples (defined as GFR>60). We used a linear regression model and controlled for batch, bisulfite conversion efficiency, gender, race, diabetes, and hypertension. This model only identified 2 probes passing Bonferroni correction (p-value < 1.19e-7) and 8 significant probes using a FDR< 0.05 cutoff for significance. We attempted to replicate the 8 significant probes in our replication dataset and were able to replicate six probes with directional consistency of the methylation change. None of these six probes were included in our original analysis top probes or the improved analysis top probes (65 probes associated with fibrosis, or the 471 probes that improved the CKD progression model). This may be due to the fact that we used age as a co-variate in the regression model. Given that age associated probes did not overlap with the fibrosis associated probes we did not think age was a major contributor of methylation changes.

We would like to point out that the current study cannot address the origin of methylation changes. We provided several alternative hypotheses, such as developmental and environmental origin based on clinical observational study. Our work is simply wanted to understand whether significant methylation differences can be observed in control and CKD samples. Future studies shall define origin and the consequence of methylation differences.

5) Could the authors provide more details as to how the enrichment within enhancers was calculated? Low levels of variation in promoters is expected as these CpG-dense regions are predominately non-variant (7) – was this accounted for?

Enrichment was calculated as an odds ratio. We evaluated the location of our top probes (either probes that significantly associated with fibrosis or improved the CKD progression model). We compared the number of top probes located in the kidney enhancer region vs. all 450k array probes located in the kidney enhancer region. For example, in evaluating the 279 top probes associated with interstitial fibrosis, we found 46 probes were located in the kidney enhancer region. We compared this to all probes on the 450K array that were used in the analysis (418339 in our original analysis). Of the 418339 probes used in the analysis, 18844 probes were located in the kidney enhancer region. In comparing this we calculated an Odds ratio 3.66 (95% confidence interval 2.62-5.02) with a p value of 2.23E-12. We compared with the background probes in order to account for the design of the 450K array probe locations. In the revised manuscript, we recalculated the odds ratios by adjusting to the reduction of the number of analyzed probes after using the Gap Hunter program (original 418,339 to 321,473).

6) 279 probes were identified as enriched for kidney enhancer regions. Whilst enhancers are significantly more tissue-specific than promoters, was this result definitively specific to kidney tissue? - or are these probes also enriched for tissue common enhancer regions - as can be evaluated in the other available Encode ChromHMM Segmentation data, for example?

To address this issue, we have downloaded regulatory annotation data for other cell types and organs from the Roadmap Epigenomics dataset. The 65 probes that associated with kidney disease development and were replicated in the second dataset, showed strong enrichment for kidney specific enhancer regions. Median fold change was 4.5 (permutation p-value 0.000299). While these probes were significantly enriched on kidney enhancers, the enrichment on other tissue enhancer

regions was lower (Supplementary Figure 4).

The results were similar when 471 probes that show association with CKD progression are analyzed. Our 471 probes showed strong enrichment for kidney specific enhancer regions (median fold change was 2.47, pvalue = 9.999e-05). (Figure 4d).

7) Can the authors include more details as to the expression analysis and how multiple testing was accounted for? If a random permutation of these data is performed, how many significant associated expression results are identified?

We used a random permutation method in order to determine the p-value cut-off for CpG methylation probes nearby (within 500kb) genes. We used a modified method previously described by Shi et al in determining cis-meQTL⁴. We determined the p-value = $8e-5$. Using this p-value cutoff in our sample size, $\beta > 0.5075$ is required to detect "methylation-expression pairs" with power > 0.8 . We identified 2798 significant "methylation-expression pairs". After removal of probes affected by genetic variation ("gap probes"), we were left with 1791 "methylation-expression pairs". Of note, a methylation probe may be nearby more than one gene (Supplementary Table 6).

Minor

- 1) Introduction pg 4: Sentence "since the epigenome is under... " – requires rewriting
- 2) Introduction pg 5: missing "to" - "...failed to pass..."
- 3) Results pg 6/7: "bisulphite conversion efficient" not "methylation conversion efficiency" as it is unmethylated cytosines that are converted
- 4) Results pg 7: "... histone tail modifications ..."
- 5) Methods pg 16: "bisulphite" not "bisulphate"

Thank you. We will correct these oversights as advised.

1. Paul DS, Teschendorff AE, Dang MA, Lowe R, Hawa MI, Ecker S, et al. Increased DNA methylation variability in type 1 diabetes across three immune effector cell types. *Nat Commun.* 2016;7:13555.
2. Andrews SV, Ladd-Acosta C, Feinberg AP, Hansen KD, Fallin MD. "Gap hunting" to characterize clustered probe signals in Illumina methylation array data. *Epigenetics & chromatin.* 2016;9:56.
3. Lunnon K, Smith R, Hannon E, De Jager PL, Srivastava G, Volta M, et al. Methylomic profiling implicates cortical deregulation of ANK1 in Alzheimer's disease. *Nat Neurosci.* 2014;17(9):1164-70.
4. Issa JP. Aging and epigenetic drift: a vicious cycle. *J Clin Invest.* 2014;124(1):24-9.
5. Teschendorff AE, Menon U, Gentry-Maharaj A, Ramus SJ, Weisenberger DJ, Shen H, et al. Age-dependent DNA methylation of genes that are suppressed in stem cells is a hallmark of cancer. *Genome Res.* 2010;20(4):440-6.
6. Rakyan VK, Down TA, Maslau S, Andrew T, Yang TP, Beyan H, et al. Human aging-associated DNA hypermethylation occurs preferentially at bivalent chromatin domains. *Genome Res.* 2010;20(4):434-9.

7. Ziller MJ, Gu H, Muller F, Donaghey J, Tsai LT, Kohlbacher O, et al. Charting a dynamic DNA methylation landscape of the human genome. *Nature*. 2013;500(7463):477-81.

□□ Reviewer #2 - expert in DNA methylation and diabetes (Remarks to the Author): □□ Clearly a study that embarks on distinguishing molecular signatures from human kidney tubule tissues is to be applauded. Rather than focus on single genes the authors discuss in the introduction the multifactorial contribution of environment and genes working in pathological states. This paper is the beginnings of a significant study that contributes to the establishment of molecular diagnostic criteria for the classification of diabetic kidney disease (DKD). Kidney function decline constitute a major proportion of intractable diabetic complications and the classification and diagnosis of DKD remains a clinical challenge. Gluck et al use array-based technologies to derive DNA methylation and gene expression using surgically resected kidney tubule tissue from human subjects with and without diabetes. Replication cohorts include kidney tubule tissues as well as blood derived methylation assessments. The major finding of this paper is the identification of specific DNA methylation □ signatures that could be used to predict kidney function decline. In striking contrast, differences in gene expression did not match with DKD subtypes and were likely driven by unknown factors that are not explored in any detail. DNA methylation is unsurprisingly a stable biological mark when compared to mRNA derived assays. The implications of these findings are unclear. The paper begins to add differential DNA methylation signatures as a potential molecular classification for DKD. □□ From a technical point of view the study was performed using 450K array and is limited to predetermined probe sets as discussed by the authors. More significant here and a point of confusion for the Nature Commun readership is the approach used is not “genome wide” methylation assessment. There is also confusion and lack in clarity in the analytical approaches used which are of concern. And while my enthusiasm is high for the article there is a paucity in mechanistic discovery. The methylomes assessments are observational and descriptive. And while there is appreciation that gene contribution is likely to be multifactorial few genes stand out as strong methyl-biomarkers of the study. □□

The paper could further be strengthened by considering the following: □□

1. While it is helpful to discuss potential mechanisms implicated in DKD that may translate into differential DNA methylation changes, what are the factors that influence DNA methylation in kidney disease? This is central to this study. Could DNA methylation be regulating pathways central to kidney disease and how could this information be used to improve predictive utility?

We would like to thank the reviewer for this important question. The goal of the study was to understand whether we can identify epigenetic changes in microdissected tubule samples in patients with diabetic kidney disease. Clinical and epidemiological studies provide a strong rationale for us to believe that epigenetic changes could be observed and contribute to disease development. A large body of literature suggests that in utero

programming could be important for kidney disease and hypertension development⁵⁻⁹, furthermore studies from the DCCT indicate that poor glycemic control could play important role in diabetic kidney disease development¹⁰⁻¹². Epigenetic changes have been reported in surrogate cell types but not in disease relevant tissue, such as human kidney samples. Defining methylation differences in kidney samples have been the central goal of the current work. Once epigenetic changes are reliable and reproducibly defined in human kidney tissue samples future studies can be developed to understand these important questions, but these studies are clearly beyond the current manuscript. A key finding of the work is that methylation can improve the precision of the prediction of kidney function decline, indicating that methylation could play important functional role in disease development. Methylation changes associated with genes of multiple pathway (Supplementary table 9) proposed to play important role in disease development, including development, signaling adhesion and immune system processes. Establishing disease causality is again beyond the scope of the current manuscript. We are in the process of establishing a CrisprCas9 system fused with Dnmt or Tet proteins that will allow us to perform site specific cytosine modification studies to define the role of specific cytosine methylation changes.

2. The authors are encouraged to examine whether differential DNA methylation patterns could also be used as a biomarker for the predictive classification of DKD (i.e. tubules) and from blood samples. Are the mechanisms that drive DNA methylation in kidney and blood generalizable? The current study does not address mechanism which would be a significant advance in the field. However, this is not the only major criticism of the study and the comments below address some of the issues. □ □

DNA cytosine methylation did improve our model for CKD progression. In this sense, methylation levels at these locations can be used as a biomarker of CKD progression (the slope of GFR decline). It is possible that there are shared mechanisms that drive DNA cytosine methylation changes in multiple cell types. Shared cell type specific and cell type independent methylation changes have also been identified in other disease conditions such as obesity and Alzheimer's disease etc¹⁻³. It seems that changes in disease relevant tissue types are more pronounced compared to surrogate cell types such as blood, but we did not perform detailed side by side comparisons. We plan to perform such studies in the future.

Abstract: □ Authors describe “genome wide methylation” in methods, this examination is by array, which is not described and the Illumina 450 array is not considered “genome wide” which is usually reserved for CpG methylation sequencing or whole genome bisulfite seq WGBS based methods. The article requires clarity on the actual DNA methylation method used, as readers will be confused throughout the manuscript. □ □

We have clarified in the abstract as well as the body of the article that this study utilized the Illumina 450K array to examine cytosine methylation across the genome at approximately 450,000 locations. In our reading of the literature genome-wide is used for studies that cover the entire genome (such as these arrays) and whole genome

covering is used for methods such as WBS. This is similar to genetic studies where genotyping arrays are also called genome wide whole WGS is whole genome covering.

Methods abstract, the authors describe gene expression and histone maps for functional integration, this requires a cleaner description, as it does refer to ENCODE datasets assessed for epigenetic changes, not necessarily derived from the study. This description needs to read more accurate as it implies further epigenetic assessments and analyses were derived from the tissues.□

Thank you we have clarified that the human kidney data was from the Roadmap epigenetics project.

Introduction:□Page 4 in the 3rd paragraph, the authors discuss intrauterine exposure and kidney disease development. The link here with development is unconvincing. Are the authors implying intrauterine DNA methylation is associated with DKD? The articles cited do not conclusively illustrate this point.□

Thank you for the note. As we discussed above, the goal of this study was to identify methylation difference in DKD/CKD samples. We cited references that potentially support this notion. Once we identify significant and consistent epigenetic changes we will have the opportunity to understand the origin of these changes such developmental or hyperglycemia induced.

Methods/Design:□□

How were the independent cohort selected? were they from different centres? and how many years apart? How is the blood replication cohort relevant when considering these samples are from American Indians? This is not only confusing but also raises other salient questions regarding the generalizability of the DKD methylation marks from the different cohorts.

This was a community based study. Participants were not selected based on exclusion or inclusion criteria but rather samples were collected during partial/full nephrectomy in accordance with protocols set forth by the TCGA. The primary and secondary kidney cohorts were selected from these samples with special attention to have a mixture of controls and cases based on GFR cutoff of 60ml/min/1.73m². The ratio of disease to control (approximately 1:3) was designed to increase the power to detect differential methylation. In the primary cohort, subjects with GFR < 60 were confirmed to have DKD on histological analysis. There was otherwise no a priori selection criteria for these samples. The secondary cohort, was designed to be a mixed etiology CKD cohort. The primary cohort was run at a separate facility than the secondary cohort, in order to control for the batch effects inherent to running the Illumina Infinium 450K array, we analyzed and normalized these cohorts separately. In addition, we ran 23 technical replicates between the primary and secondary cohorts to better assess this batch effect.

The blood replication cohort is the only available cohort from patients with diabetic kidney disease with both 450K methylation data and longitudinal eGFR data. It is true

that this is a different population with a different genetic background than ours. However, these issues should have biased us towards the null. In contrast, despite this unlikely overlap, we were able to replicate the correlation between methylation changes and kidney disease.

Is there any genetic data available for the samples used in this study? Is it possible that genetic heterogeneity due to population genetic variation in CpG-SNPs could be driving methylation differences?

We took several steps to remove genetic heterogeneity. Originally, we removed methylation probes at the location of SNPs (dbSNP137) or within 1 bp extension in our original analysis. In our revised analysis, as suggested by reviewer 1, we have applied the Gap Hunter method to remove any potential probes influenced by genetic variation. Please see above discussion.

A combination of Bonferroni and FDR correction was used for Multiple comparisons throughout the study. For consistency in statistics can the authors use one or the other? The paper does not estimate what statistics were used for correction.

Thank you for pointing this out. In the revised manuscript, we consistently used the false discovery rate method (FDR < 0.05) to define statistical significance.

Did the authors perform any power analysis prior to the DNA methylation analysis?

Power analysis for EWAS projects are difficult. We used methods described by Tsai et al. for power calculation (utilizing both t-tests and Wilcoxon signed rank test) taking the distribution of Beta into account and using case/control analysis for various sample sizes. Based on their estimations, for our sample size (22 pairs of case/control in our primary data set and 37 pairs of case/controls in our replication data set), we have 80% power to determine an approximately 20-25% mean difference in methylation at genome-wide significance. In addition, our power is increased because we have >3:1 ratio of controls to cases.

No methyl-validation of probes were attempted and the study requires stronger assessment of methylation difference using an independent assay. The differences in the methylation levels of the CG probes could be a function of the array.

For one top probe, we completed experiments to validate the methylation changes as measured by the Illumina Infinium 450K arrays. For 3 samples with DKD and 4 control samples, we microdissected the human kidney samples and isolated the tubule compartment DNA. DNA was bisulfite converted, amplified with PCR, and transformed into bacteria. 15 colonies were selected per sample and the PCR segment was sequenced. Bisulfite converted sequences were compared with genomic DNA sequence using QUMA: quantification tool for methylation analysis¹³. Correlation coefficient between Illumina Infinium 450k array beta value and percent measured methylation for this locus was 0.88 (p-value = 0.0083).

However, it is worth noting that for samples with a smaller degree of methylation change, this sample size is insufficiently powered to assess this change. Others have shown that the Illumina Infinium 450k arrays have good technical replications¹⁴.

Results:

The main research question for this manuscript is to identify cytosine methylation difference in patients with diabetic kidney disease, how did the authors adjust for diabetes?

Diabetes at the time of human kidney sample collection was included in our data set. In all linear regression analysis, diabetes status (yes or no) was included as a factor variable.

Could the authors assess the effect of gender, age and race on the differentially methylated probes?

The data was adjusted for gender, age and race. As detailed above, we ran a linear regression to identify age associated probes. Please see description above. In summary, few probes showed statistically significant association with age and they were different from the fibrosis associated probes. Sex chromosome specific probes were removed from the analysis. Overall our dataset is too small to identify secondary outcomes such as race, age and gender.

Odds ratio analysis was used to define functional importance of methylation differences. ChromHMM uses background data to calculate overrepresented in differential data, can the authors give details on the background dataset used?

As described above, enrichment was calculated as an odds ratio. We evaluated the location of our top probes (either probes that significantly associated with fibrosis or improved the CKD progression model). We compared the number of top probes located in the kidney enhancer region vs. all 450k array probes located in the kidney enhancer

region. For example, in evaluating the 279 top probes associated with interstitial fibrosis, we found 46 probes were located in the kidney enhancer region. We compared with all probes on the 450K array that were used in the analysis (418,339 in our original analysis). Of the 418,339 probes used in the analysis, 18,844 probes were located in the kidney enhancer region. In comparing this we calculated an Odds ratio 3.66 (95% confidence interval 2.62-5.02) with a p value of 2.23E-12. We compared with the background probes in order to account for the design of the 450K array probe locations.

In the revised manuscript, we also applied the Gap Hunter method to account for genetic variation and large number of probes were removed. The odds ratios were recalculated for the revised manuscript.

Gene expression was also performed in (n=58), however the results are not shown. One of the aims of the study was integrated methylation changes with regulatory and gene expression changes, yet there are no results shown. Authors only show gene expression of a few selected genes. If this was not the aim of the study than it remains confusing what hypothesis the authors are assessing. □ □

Differential gene expression analysis was not the main goal of this work. The Susztak group has extensively published on differential gene expression in human diabetic and CKD samples (Beckerman eBioMedicine 2017, Kang et al Nature Medicine 2015)^{15,16}. In this study, gene expression data was used to understand whether methylation changes correlate with gene expression changes, such correlation in our view could highlight functionally important methylation changes. We had 77 samples with both methylation and gene expression data. We used a random permutation method in order to determine the p-value cut-off for CpG methylation probes nearby (within 500kb) genes. We used a modified method previously described by Shi et al in determining cis-meQTL⁴. We determined the p-value = 8e-5. Using this p-value cutoff in our sample size, beta>0.5075 is required to detect "methylation-expression pairs" with power>0.8. We identified 2,798 significant "methylation-expression pairs". After removal of probes affected by genetic variation ("gap probes"), we were left with 1791 "methylation-expression pairs". Of note, a methylation probe may be nearby more than one gene (Supplementary Table 6). Since we were interested in methylation-expression pairs where the methylation occurred in kidney gene regulatory region, we narrowed our results to these "pairs". This data was presented in table 3, table 5 and supplementary table 10.). We have also included as supplementary table 5 and supplementary table 9, respectively, the DAVID pathway analysis for the top methylation probes associated with degree of interstitial fibrosis and top methylation probes that improve our model for CKD progression.

Page 7, how was linear regression adjustments made for age, sex, race, diabetes, hypertension and batch effect? Description of the methylation conversion efficiency is also warranted showing how this was performed. □

Using the linear regression function in R, we compared methylation level (M value) of each probe to the outcome of interest (for example, interstitial fibrosis) while adjusting for age, sex, race, diabetes, hypertension, batch effect, bisulfite conversion efficiency, and lymphocytic infiltrate (sex, race, diabetes, hypertension, and batch were all treated as factor variables). The bisulfite conversion efficiency was calculated using the bisulfite conversion control probes, based on Illumina guidelines. Ten CpG sites designated by Illumina as control sites (6 CpGs targeted by type I probes and 4 CpGs targeted by type II probes), where we expect each CpG to be 100% methylated, are used to control for non-complete bisulfite conversion. The bisulfite conversion efficiency was used in the primary analysis is the median methylation estimate from the ten control sites. The bisulfite conversion was calculated by taking the median value of the probes that Illumina provides to estimate bisulfite conversion efficiency¹⁷.

Page 7, “to better control for batch effect” this is unusual and it’s unclear why this was not performed earlier in the paper. □ □ Page 7, ChromHMM integration performed on datasets derived for the histone modifications in “human kidney” presumably they mean non-DKD tissue? If so, how are the epigenetic comparisons made for diabetic and non-diabetic kidneys? □

Thank you for pointing this out, we have clarified this statement. All regression analysis was controlled for batch effect as defined by the Illumina slide (Sentrix ID). However, our primary kidney cohort and secondary kidney cohort were run at different locations at different times. Since we were unsure how location and time would affect our samples, we first ran the primary and secondary cohort separately and normalized these results separately. However, we then combined the primary and secondary cohorts (and normalized them together). Since we were able to replicate the 65 significant probes, we felt that these probes were most reliably associated with interstitial fibrosis.

Yes, the ChromHMM integration was performed on non-DKD tissue. The diabetic and fibrosis state affects the cell composition of the kidney and will also likely influence the histone modification epigenome maps.

Page 7, the first description of the array is in the results section of the paper. □ □

Thank you for bringing this to our attention, we have described the array in the abstract and introduction.

Page 8, the paper needs to define the “stringent statistical criteria” used for methylation and gene expression changes. What are they and better descriptions required to justify methylation differences. □ □

We used a random permutation method in order to determine the p-value cut-off for CpG methylation probes nearby (within 500kb) genes. We used a modified method previously described by Shi et al in determining cis-meQTL⁴. We determined the p-value = $8e-5$. Using this p-value cutoff in our sample size, $\beta > 0.5075$ is required to detect

"methylation-expression pairs" with power>0.8. We identified 2798 significant "methylation-expression pairs". After removal of probes affected by genetic variation ("gap probes"), we were left with 1791 "methylation-expression pairs". Of note, a methylation probe may be nearby more than one gene (Supplementary Table 6). We added this description to the manuscript.

Page 8, methylation is inversely correlated for HOPX gene expression. The importance of this finding in the paper is not strong. What available data is there that HOPX is regulated by DNA methylation driving gene expression?

Understanding the functional role of methylation changes was not the primary goal of the current study. We correlated the methylation changes with gene expression changes to highlight likely causal methylation changes. We identified methylation and gene expression changes in the HOPX gene, but establishing the role of HOPX in kidney disease development is clearly beyond the scope of the current manuscript. Of note, after controlling for probes under genetic influence ("gap probes") the methylation probe associated with HOPX expression changes was removed from our analysis and therefore this is no longer presented as a top methylation locus.

Page 9, "the methylation level of 1,131 probes was significantly associated with renal function decline..." unclear in which cohort the result is observed. □□

This was observed in the subset of the primary cohort with longitudinal eGFR measurements (n=69). We have clarified this in the manuscript.

Discussion:□□

Page 12, "robustness including blood sample from the DCCT study" ... Yet the overlap between data generated in this study and DCCT is limited to 2 CpG sites (page 11). The paper does not illustrate robustness of the association, rather, observation the 450K array identifies methylation differences at probes that might be considered passing statistical testing.□

Thank you for this comment. We have appropriately qualified this replication. □

Page 12, it remains unclear how the methylation changes for the genes screened for methylation difference, such as HOPX are functionally important.□□

We hypothesize that methylation changes, especially those located in organ specific active gene regulatory regions (such as promoters and enhancers) are functionally important in altering gene expression patterns. This is an exploratory data set, further testing would be needed to establish causality, for example in a mouse model. We have clarified this in our conclusion.

Figures/Tables:□□

Table 1, Diabetes duration not shown. Over how many years and measurements was eGFR assessed? □□

Our cases were mostly from patients with type2 diabetes as this was a community cohort study. As with most subjects the duration of type2 diabetes is less clear. Longitudinal eGFR data was available for 69 subjects. Mean timespan was 2.4 years, median timespan was 2 years, and standard deviation was 1.5 years. We have added this to table 1.

Table 3, can authors report effect size of methylation change for the probes shown? We have added the coefficient estimates from the linear regression for each methylation probe shown.

Figure 3a and 4a, can authors show confidence intervals on plots? □□
OR with 95% confidence intervals were shown. However, this may be better displayed as a Forrest plot therefore we changed the figure to better depict this information.

No description table for gene expression cohort is shown. □□
We have provided a supplementary table 3 with demographic and clinical characteristics of the sub-cohorts (sub-cohort with gene expression data and sub-cohort with longitudinal eGFR data).

Suppl. Information: □□

The paper should show methylation data from all samples using PCA plots in order for the reader to visualize the variability in methyl-signals.

We have included a PCA plot for each cohort as supplementary figures 1, 3, and 10.

□□□□Reviewer #3 - expert in diabetic nephropathy (Remarks to the Author):□

□ In this retrospective, observational analysis Gluck and coworkers tried to define the wide cytosine methylation differences in microdissected human kidney tubule epithelial cells of patients with diabetes and kidney disease and to evaluate whether and to what extent analysis of cytosine methylation levels can improve the predictive value of current models of renal function decline. The issue is of potential methodological and clinical interest. Unfortunately, the study is flawed by major limitations in patient selection and outcome analyses, the analyses fail to address the primary question of the paper, data are unclear and their presentation is confusing and potentially misleading. □□

More specifically: □□

1. The study population is poorly characterized. Main clinical and laboratory characteristics at the time the kidney tissue was sampled should be provided in Table 1, including detailed information about factors that may affect renal disease outcome such

as blood pressure and metabolic control, serum lipids, underlying histological diagnosis of kidney disease, concomitant treatment with drugs such as ACE inhibitors, ARBs, HMGCoA inhibitors, blood glucose and blood pressure lowering agents and others that may affect GFR decline over time (the primary outcome of the study). All these parameters should be considered in the univariable and multivariable models used by the Authors. □ □

Thank you for pointing this out. We provide a comprehensive clinical and histopathological information for the samples. We have additional clinical information for some of the samples. However, in our review the literature the role of lipids in DKD development is less well established. We did not have reliable time adjusted information available for all our samples. Our sample size of 69 (for the longitudinal analysis) would also not allow adequately powered subgroup analysis for such factors. In general, these factors have not been shown to improve GFR decline models in the past.

2. The title and data presentation throughout the text are misleading. Why the Authors focus the attention on diabetic kidney disease when only 41 of the 91 study patients were diabetics? □ □

We needed to use control samples for the analysis. Having non-diabetic samples allowed us to control for the effect of diabetes. Differentially methylated loci can only be ascertained by comparison of disease to controls (if we only examined disease then we would have no reference for comparison). Therefore, the main outcome of our analysis was comparing patients with diabetic kidney disease with controls and differences between these populations may point to underlying pathophysiologic mechanisms or biomarkers of disease.

3. Again, the Authors state that methylation differences with genome wide significance can be detected in kidney tubule samples of patients with diabetic kidney disease. Thus, why data on 50 non-diabetic patients are reported in Table 1? Information about non diabetic patients should be deleted.

□ □

Since there is no universal reference for normal kidney methylome, we would be unable to detect differences in methylation without these control samples for comparison. By using approximately 3:1 control to disease population we also increase our power to detect differences in methylation patterns.

4. Independent of the above, the Authors should acknowledge that the study population is small, which may affect the power of the analyses and the robustness of the findings. In this context, a justification for the sample size should be provided to ensure that study findings are not casual. □

The sample size is as large or larger than other sample sizes used to analyze methylome data (specifically using the Illumina Infinium 450k BeadChip) in patients with kidney disease. By using 3:1 control to disease we also increase our power to detect changes. Power analysis for EWAS projects are difficult to perform. We used methods

described by Tsai et al. estimate power utilizing both t-tests and Wilcoxon signed rank test and taking the distribution of Beta into account and using case/control analysis for various sample sizes. Based on their estimations, for our sample size (22 pairs of case/control in our primary data set and 37 pairs of case/controls in our replication data set), we have 80% power to identify an approximately 20-25% mean difference in methylation at genome-wide significance.

5. The authors do not take into consideration that type 1 and type 2 diabetes are two different disease entities, with different etiologies, phenotypes, outcomes and treatment. In type 1 diabetes hyperglycemia is the direct consequence of impaired insulin production, in type 2 diabetes hyperglycemia is just one of the several manifestations of the metabolic syndrome, including hypertension, dyslipidemia, obesity, insulin resistance and other. Thus, the Authors should clarify how many of their patients had type 1 or type 2 diabetes and should consider them separately. □

We appreciate and agree with the comment however, we are not aware of clinical studies that would have been able to differentiate diabetic kidney disease in patients with type1 from type 2 diabetic patients. The gold standard histological diagnosis of DKD is basement membrane thickening, mesangial expansion, nodular sclerosis and interstitial fibrosis is used for both type1 and type2 subjects. While we did not have very clear information on the type of diabetes, almost all of our subjects were presumed to have type2 diabetes as this was a community-based cohort. Determining the type of diabetes an exceedingly difficult task that requires special antibody detection for type1 diabetes amongst other parameters.

□6. The definition of diabetic kidney disease is also nebulous. In patients with type 2 diabetes, kidney changes may include typical focal glomerular sclerosis (in a minority of cases), nephroangiosclerosis, ischemic kidney disease, tubule-interstitial disease, concomitant primary glomerular diseases, ageing-related changes and other changes that can be observed in different combinations in different patients. Probably these different histological patterns reflect different pathogenic mechanisms. Thus it is hard to believe that a single predictive model may predict disease outcome to a similar extent in such a heterogeneous population of patients. This is an issue that should be taken in due consideration by the Authors and should be adequately discussed. In table 2 the authors should report the histological diagnosis of the study patients. □□

The pathomechanism of DKD is not fully understood. We used the gold standard diagnosis for DKD, which is based on histological evaluation of the kidney. Our study aimed to identify consistent methylation changes in patients with DKD. We correlated the degree of tubulointerstitial fibrosis with methylation changes to account for disease severity. Fibrosis as a common mechanism of kidney disease and strong factor determining progression was a primary outcome of the study.

7. More in general no information is provided about the selection criteria for study participation. According to data in Table 1 it is conceivable that a subgroup of "healthy" subjects without diabetes, hypertension or proteinuria was also included. This should be

clarified. How can methylation changes in “healthy” subjects predict the outcome of diabetic kidney disease? Incidentally, reasons for nephrectomy should be reported.

□□

Thank you for the note. Our study was a community-based study, we included everyone who had undergone nephrectomy for renal cell cancer in our institution. This was a community based study. Participants were not selected based on exclusion or inclusion criteria but rather samples were collected during partial/full nephrectomy in accordance with protocols set forth by the TCGA. The primary and secondary kidney cohorts were selected from these samples with special attention to have a mixture of controls and cases based on GFR cutoff of 60ml/min/1.73m². As discussed above control (healthy) subjects were used to define differences between control and DKD subjects. The ratio of disease to control (approximately 1:3) was designed to increase the power to detect differential methylation. In the primary cohort, subjects with GFR < 60 were confirmed to have DKD on histological analysis. There was otherwise no a priori selection criteria for these samples.

8. More detailed information should be provided about outcome analyses. How eGFR slopes were calculated? Did the authors account for the acute changes associated with nephrectomy (sharp GFR reduction after nephrectomy followed by compensatory hyperfiltration of residual tissue) and may be subsequent decline due to exhaustion of surviving nephrons? Which was the minimum number of GFR estimations and the minimum follow up required for slope calculations?□

Thank you for asking this important question. Subject specific unadjusted eGFR slopes were determined by linear regression across all available eGFR measures. Only subjects with a minimum of 3 eGFR estimations and longitudinal eGFR measurements for at least three months post nephrectomy were included for the analysis. Three months post-nephrectomy was chosen as a minimum as a way to minimize the acute changes peri-nephrectomy. The mean timespan of follow-up was 2.4years (standard deviation =1.5 years). Subjects with unadjusted eGFR slope < -40 or > 40 ml/min/1.73m²/year were excluded.

In order to account for random variation as well as non-normal distribution of the data, subject specific eGFR slope was adjusted using a form of mixed effect model, best linear unbiased predictor (BLUP). Subject specific adjusted eGFR slope and variance were used in weighted regression analysis (weight = inverse of the variance of the slope) such that subjects with increased variability to their eGFR slope were weighted less in the analysis.

The rate of GFR decline in our cohort was similar to other studies reported for the degree of kidney fibrosis. Most cases are obtained from partial nephrectomies as total nephrectomy is hardly ever performed for renal cell cancer. In addition, changes observed in the kidney cohort were confirmed in the Pima cohort that is an extremely well phenotyped diabetic cohort, without cancer or partial nephrectomy.

9. The study design should be described in title and abstract; study setting, participating

centers, recruitment period, data monitoring should be described in detail in the text. A justification of the sample size should be provided. Was an informed consent required for the use of kidney tissue for the purposes of the study? Was the protocol approved by an ethical committee?

We have clarified the study design in the abstract. This study utilized the healthy portion of full/partial nephrectomy samples that were collected in accordance with protocols established and standardized by The Cancer Genome Atlas (TCGA) project. This project was deemed exempt by the Internal Review Board at the University of Pennsylvania as only de-identified kidney samples were used for the analysis. IRB approval was obtained both at University of Pennsylvania and at the Albert Einstein College of Medicine.

10. Although I recognize the difficulty in finding biopsy samples, the validation step for the methylation changes that predict kidney function decline in peripheral blood mononuclear cells (PBMCs), instead of microdissected tubules, is not fully convincing. Indeed, one of the first assumptions is that methylation is cell type-specific. What is more, one of the top probes found to correlate with kidney function decline is not confirmed by results published by Chen et al. on whole blood from patients treated with conventional glycemic therapy and diabetic complications. To use PBMCs in the validation step, the authors should confirm that microdissected tubule methylomes match (fit) with the PBMCs ones in the primary and replication cohorts.

Thank you for pointing this out. There is a large body of literature on using surrogate cell types to define epigenetic changes that are observed in disease relevant cell types. For example, in the Alzheimer literature some methylation changes observed in blood samples correlate with brain specific changes^{1,2}. Similar observations have been published in the obesity literature³. It seems that changes on surrogate cell types are less pronounced, similar to our observations. We did not have matching blood samples from our study to perform a paired kidney, PBMC analysis. We also would like to point out that the Chen dataset obtained from PBMCs is very small (n= 63). Upon adjusting for multiple comparisons (the way we did in the current project) no probe passed genome wide significance in the publication by Chen et al. Rather, the work used fold changes (FCs) ≥ 1.3 between cases and controls and nominal $P < 0.005$ for outcome. For this reason, we are less surprised that we were unable to validate our results in this dataset. The two probes that we had originally replicated in this cohort were removed by the “gap hunter” program that removes probes with likely variation due to genetic background. We have delete this section from the revised manuscript.

11. Locus specific validation of kidney cytosine methylation and gene expression changes for the top replicated probes associated with interstitial fibrosis and kidney function decline are lacking and should be provided through bisulfite sequencing and real time qPCR experiments, respectively.

As described above, for one top probe, we completed experiments to validate the methylation changes as measured by the Illumina Infinium 450K arrays. For 3 samples with DKD and 4 control samples, we microdissected the human kidney samples and isolated the tubule compartment DNA. DNA was bisulfite converted, amplified with PCR, and transformed into bacteria. 15 colonies were selected per sample and the PCR segment was sequenced. Bisulfite converted sequences were compared with genomic DNA sequence using QUMA: quantification tool for methylation analysis¹³. Correlation coefficient between Illumina Infinium 450k array beta value and percent measure methylation for this loci was 0.88 (p-value = 0.0083).

However, it is worth noting that for samples with a smaller degree of methylation change, this sample size is insufficiently powered to assess this change. Others have shown that the Illumina Infinium 450k arrays have good technical replications¹⁴.

12. The authors identified 7 probes associated with interstitial fibrosis, but described only the one correlated with HOPX transcript level. It is not clear why they omitted the other descriptions. The authors should discuss the possible involvement of all the differentially methylated and expressed genes (listed in Table 3) in the progression of interstitial fibrosis in DKD. Moreover, in vitro experiments in proximal tubule cells exposed to high glucose or silencing/overexpressing HOPX are needed to elucidate the involvement of HOPX in interstitial fibrosis. □□

We included in our discussion a description of all top methylation-expression pairs identified. We would like to emphasize that the goal of the current study was to identify consistent and reproducible methylation changes in patients with kidney disease. Further research is needed to investigate the role of cytosine methylation changes and specific gene expression changes in diabetic kidney disease.

13. Similarly to point 2, the authors should describe other possible candidates (besides EGF) as predictors/biomarkers for renal function decline, focusing on those that are known to play a role in kidney patho-physiology (for example collagen, as shown in

Supplementary Table 6). □□

We included a broader discussion of other methylation-expression pairs that improved our progression model as well as gene ontology (supplementary table 9).

14. The abstract results are misleading. It seems like methylation changes in the kidney promoter regions next to EGF were validated in blood samples of the independent cohort, while they were not. □□

We clarified that only a subset of probes that improved the progression model were replicated in the blood replication cohort.

Minor points: □□

- a) Several inaccuracies occur throughout the text: - Is the replication cohort left with 416354 CpG probes (as stated in the Procedures section) or 406354 (correct) (as stated in Figure 2) .

We apologize for this typo. Indeed 406,354 was correct, however this number is now different in the updated analysis, due to removal of the probes recommended by the Gap Hunter method.

- In the last paragraph on page 13, the authors first assert that subjects underwent full/partial nephrectomy, but then that all subjects underwent partial nephrectomy. Please correct accordingly. □

We have clarified this.

- In the abstract, the authors should substitute diabetes with diabetic kidney disease (page 2, line 9)□.

We left this as is since we also had control patients with diabetes but no kidney disease.

- On page 10, line 24, the authors reported 73 CpG probes, while in Figure 5 and its legend, 76 are reported (correct). Please correct accordingly. □□

We apologize for this typo. Indeed 76 was correct, however this number is now different in the updated analysis.

b) Results and Procedures should be divided into paragraphs and titled in order to facilitate reading of the paper. □

Thank you for this suggestion, we have added titles to help guide the reader. □

c) The authors should specify in greater detail which Affymetrix microarray they used. Affymetrix U133A arrays – we have added this detail to the manuscript.

Additional References

- 1 Hannon, E., Lunnon, K., Schalkwyk, L. & Mill, J. Interindividual methylomic variation across blood, cortex, and cerebellum: implications for epigenetic studies of

- neurological and neuropsychiatric phenotypes. *Epigenetics* **10**, 1024-1032, doi:10.1080/15592294.2015.1100786 (2015).
- 2 Klein, H. U., Bennett, D. A. & De Jager, P. L. The epigenome in Alzheimer's disease: current state and approaches for a new path to gene discovery and understanding disease mechanism. *Acta Neuropathol* **132**, 503-514, doi:10.1007/s00401-016-1612-7 (2016).
- 3 Wahl, S. *et al.* Epigenome-wide association study of body mass index, and the adverse outcomes of adiposity. *Nature* **541**, 81-86, doi:10.1038/nature20784 (2017).
- 4 Shi, J. *et al.* Characterizing the genetic basis of methylome diversity in histologically normal human lung tissue. *Nat Commun* **5**, 3365, doi:10.1038/ncomms4365 (2014).
- 5 Barker, D. J., Osmond, C., Golding, J., Kuh, D. & Wadsworth, M. E. Growth in utero, blood pressure in childhood and adult life, and mortality from cardiovascular disease. *BMJ* **298**, 564-567 (1989).
- 6 Chen, P. *et al.* Differential methylation of genes in individuals exposed to maternal diabetes in utero. *Diabetologia* **60**, 645-655, doi:10.1007/s00125-016-4203-1 (2017).
- 7 West, N. A., Kechris, K. & Dabelea, D. Exposure to Maternal Diabetes in Utero and DNA Methylation Patterns in the Offspring. *Immunometabolism* **1**, 1-9, doi:10.2478/immun-2013-0001 (2013).
- 8 White, S. L. *et al.* Is low birth weight an antecedent of CKD in later life? A systematic review of observational studies. *Am J Kidney Dis* **54**, 248-261, doi:10.1053/j.ajkd.2008.12.042 (2009).
- 9 de Jong, F., Monuteaux, M. C., van Elburg, R. M., Gillman, M. W. & Belfort, M. B. Systematic review and meta-analysis of preterm birth and later systolic blood pressure. *Hypertension* **59**, 226-234, doi:10.1161/HYPERTENSIONAHA.111.181784 (2012).
- 10 Chen, Z. *et al.* Epigenomic profiling reveals an association between persistence of DNA methylation and metabolic memory in the DCCT/EDIC type 1 diabetes cohort. *Proc Natl Acad Sci U S A* **113**, E3002-3011, doi:10.1073/pnas.1603712113 (2016).
- 11 de Boer, I. H. *et al.* Long-term renal outcomes of patients with type 1 diabetes mellitus and microalbuminuria: an analysis of the Diabetes Control and Complications Trial/Epidemiology of Diabetes Interventions and Complications cohort. *Arch Intern Med* **171**, 412-420, doi:10.1001/archinternmed.2011.16 (2011).
- 12 Miao, F. *et al.* Evaluating the role of epigenetic histone modifications in the metabolic memory of type 1 diabetes. *Diabetes* **63**, 1748-1762, doi:10.2337/db13-1251 (2014).
- 13 Kumaki, Y., Oda, M. & Okano, M. QUMA: quantification tool for methylation analysis. *Nucleic Acids Res* **36**, W170-175, doi:10.1093/nar/gkn294 (2008).
- 14 Forest, M. *et al.* Agreement in DNA methylation levels from the Illumina 450K array across batches, tissues, and time. *Epigenetics* **13**, 19-32, doi:10.1080/15592294.2017.1411443 (2018).
- 15 Beckerman, P. *et al.* Human Kidney Tubule-Specific Gene Expression Based Dissection of Chronic Kidney Disease Traits. *EBioMedicine* **24**, 267-276, doi:10.1016/j.ebiom.2017.09.014 (2017).

- 16 Kang, H. M. *et al.* Defective fatty acid oxidation in renal tubular epithelial cells has a key role in kidney fibrosis development. *Nat Med* **21**, 37-46, doi:10.1038/nm.3762 (2015).
- 17 De Jager, P. L. *et al.* Alzheimer's disease: early alterations in brain DNA methylation at ANK1, BIN1, RHBDF2 and other loci. *Nat Neurosci* **17**, 1156-1163, doi:10.1038/nn.3786 (2014).

Reviewers' comments:

Reviewer #1 (Remarks to the Author):

Gluck *et al.* have comprehensively reviewed their manuscript on Kidney DNA methylation assessment in regard to renal function decline in diabetic kidney disease. The authors are to be commended for taking on board the issues raised and rigorously reanalysing their data in this light. Comments on their responses to my previous review points are below, including a couple of remaining issues which, if the authors could address, I think would further improve the manuscript.

- 1) As suspected, due to cell type and genetic heterogeneity, the authors have identified very significant inflation in their previously presented results. ~20% of the probes were shown to have potential genetic influence via Gap Hunting. Furthermore, inflammatory infiltration to cell type proportions was shown to be the strongest contributor. Therefore, the inclusion of a measure of lymphocytic infiltrate into the analysis model has led to a strong reduction in this inflation. Inflammatory cells must be accounted for in many tissue samples and this was also recently dissected within saliva/buccal samples by Zheng *et al.*¹ There may be some benefit in utilising an adaptation of this paper's methodology to further improve their correction for this confounding biological factor. The QQ-plots, whilst significantly improved, are still inflated post this reanalysis. This remains one of my two main concerns, that further cell-type heterogeneity, perhaps additional uncounted inflammatory or the mixture of other non-inflammatory cells in the biopsy, may still be contributing to the result. This could be further explored, as above, or at least acknowledged in the manuscript.
- 2) Yes, as detailed the authors have discussed this, as above.
- 3) The common Transcription Factors explanation detailed here does not appear to be included in the revised manuscript, unless I missed it? Furthermore, as the authors state "Our difficulty in replicating results in surrogate cell types after removal of underlying genetic variability is in keeping with observations from other diseases." Therefore, that the previous positive blood tissue results, prior to Gap Hunting probe exclusion, were enriched for confounded genetic effects. This is my second issue, in that these caveats should be directly declared. It should be stated in the paper that these surrogate tissue results were previously strongly confounded by genetic effects and that a minority of undetected genetic effects could still be present. Also, as well that common inflammatory cell changes could be contributing to the signal seen across both different tissue type samples.

- 4) Good to have explored the potential for ageing-related epigenetic changes to be contributing to this signal and excluding this possibility.
- 5) Additional details on the enrichment analysis have been provided.
- 6) Again, good to explore and define that these are explicitly kidney-specific enhancers.

7) Thanks to the authors for providing this additional detail regarding the expression analysis – and the use of the Shi *et al.* method is appropriate for this.

1. Zheng et al. (2018) *Epigenomics* 10.2217/epi-2018-0037

Reviewer #2 (Remarks to the Author):

Point 1. While it is appreciated that mechanisms underpinning DNA methylation might be beyond the scope of the study, the illumina 450k array allows the investigators to expand on the CG methylation from the signatures on the array to implicate genes and pathways that contribute to DKD. Defining the probes on the array is an important first step to the study and extending the analysis to identify hypermethylation and hypomethylation groups should identify genes containing differentially methylated sites in promoters or gene bodies using analyses such as IPA as recently shown by Chen et al (Proc Natl Acad Sci U S A. 2016 May 24;113(21):E3002-11). Therefore, based on the original review points raised, understanding what methylation connections with regulatory pathways are not necessarily outside the scope of the current study. With this information the authors should be able to assign what methylation sites and states (reduced or increased or unchanged) can be used to improve predictive utility for DKD. This is an important consideration that was not addressed and should receive more attention for the readers of this journal. In addition, Chen et al were able to show the functional control of methylation using cells in experiments exposed to low-high glucose showing gene expression was inversely correlated with changes in DNA methylation. An important note here is the methylation datasets from both monocytes and whole blood cells by Chen are informative to the 115 blood samples derived from subjects examined in the replication cohort by Gluck et al. An assessment would have been informative.

Point 2. Methylation in tubules and blood samples. The authors discuss this important pitfall in the discussion on page 15, however, readers of this journal need to see possible shared mechanisms that drive methylation in the multiple cell types. Describing what methylation states is only the beginning and the general feeling by the authors is this is beyond the study scope but should not be. Indeed, there are experiments that assess the functionality of methylation states and sites that are important in providing knowledge showing methylation is not an epiphenomenon. Chen et al provide examples of methylation changes in both WB and monocytes associated with gene expression changes.

Point 3. Given that small numbers of individuals are studied and present with varying degrees of DKD and the methylation probe cg21048700 is identified in a kidney inactive region which is thought to be "associated" with COL3A1. The link between the methylated probes and genes implicated with DKD is not strong and raises questions as to whether the methylation is beyond chance? Additional data is necessary to address this.

Abstract: the contemporary use of the terms "genome wide cytosine methylation" is confusing when attached to 450K array and this will be noticed by the broad Nature Comm readership, furthermore, this is highlighted with the technological advances of the 850k array which many do not attach "genome wide cytosine methylation". Whilst it is understood that the 450K array is a cursory

assessment of methylation, it by no means would be referred to as "genome wide" and this appears to be a misunderstanding here. It is simpler and more accurate to use "Illumina Infinium 450K arrays".

Methods/Design: The authors concede the replication cohort using blood samples is likely to be different from the kidney tissues and this could be the case for methylation. The article by Chen et al (Proc Natl Acad Sci U S A. 2016 May 24;113(21):E3002-11) show dramatic changes in DNA methylation including a range of genes and probes using the Illumina 450k array, the same method used in this article. In fact, that article is important, not only because of the relevant experimental design and 450 array approaches, but is relevant because they also identified probes from the 450k array associated with genes including TXNIP, CYB5b and many others. The dataset by Chen et al is rich with valuable information that could have strengthened the Gluck et al article. It is unclear why this relevant dataset that shows methylation in monocytes and whole blood was not carefully explored in relation to the current article. Any analysis and comparison would be useful, comparing similar technologies (450k array) and disease (diabetic complications).

Power calculations: 80% is too low for a study of the size and what appears to be replicated between blood and kidney tissue may very well reflect probe bias. This is why an independent method is useful.

Reviewer #3 (Remarks to the Author):

1. Patient groups are still poorly characterized, despite the authors' claim that they have provided a comprehensive clinical and histopathological information.

2. The authors claim that they have clarified the study design in the abstract. This is simply not true. Moreover, they did not provide any information about study setting, participating centers, recruitment period and data monitoring, as requested.

3. The authors asserted that in the primary cohort they included controls (non-diabetic CKD patients). However, this has not been specified in the manuscript, but, more importantly, no clinical data on diabetic and non-diabetic CKD patients (controls) have been provided. Data of the two

patient populations should be shown. The choice of non-diabetic CKD patients is crucial, as they must have the same degree of renal insufficiency and proteinuria as diabetic patients.

4. I recognize the step forward upon filtering out with Gap Hunter method. However, this new approach has not confirmed some probes associated with interstitial fibrosis found in the original paper's analysis, in particular as regards the one correlated with the stem cell transcription factor HOPX (Hop homeobox) levels that, according to the authors, could have a pivotal role in disease development.

5. The authors didn't specify whether the degree of cg24818418 methylation associated with EGF (Epidermal Growth Factor) transcript level, and able to predict kidney function decline, was validated in the independent cohort. This should be the most important confirmation finding of their study.

6. The authors did not provide a convincing answer about the use of surrogate cells to define epigenetic changes observed in the kidney. Two of three papers cited by the authors in their defense actually do not support their claims. Indeed, Hannon et al. found that for the majority of DNA methylation sites, interindividual variation in whole blood is not a strong predictor of interindividual variation in the brain, while Klein and co-authors asserted that "it is unlikely that there is a strong, direct correlation between the epigenomes of the brain and blood such that measuring a given epigenomic feature in blood offers a reasonable proxy for the same feature in the brain for most genomic loci".

7. The authors should properly cite the method they used to estimate the power of their sample size (Tsai et al., ?) in the main text.

Minor points:

- Supplementary tables are erroneously named. Please check and correct.
- The authors should re-write the sentence on page 8, lines 14-17 "Fold change...analysis".

Gluck et al. have comprehensively reviewed their manuscript on Kidney DNA methylation assessment in regard to renal function decline in diabetic kidney disease. The authors are to be commended for taking on board the issues raised and rigorously reanalysing their data in this light. Comments on their responses to my previous review points are below, including a couple of remaining issues which, if the authors could address, I think would further improve the manuscript.

1) As suspected, due to cell type and genetic heterogeneity, the authors have identified very significant inflation in their previously presented results. ~20% of the probes were shown to have potential genetic influence via Gap Hunting. Furthermore, inflammatory infiltration to cell type proportions was shown to be the strongest contributor. Therefore, the inclusion of a measure of lymphocytic infiltrate into the analysis model has led to a strong reduction in this inflation. Inflammatory cells must be accounted for in many tissue samples and this was also recently dissected within saliva/buccal samples by Zheng et al.¹ There may be some benefit in utilising an adaptation of this paper's methodology to further improve their correction for this confounding biological factor. The QQ-plots, whilst significantly improved, are still inflated post this reanalysis. This remains one of my two main concerns, that further cell-type heterogeneity, perhaps additional uncounted inflammatory or the mixture of other non-inflammatory cells in the biopsy, may still be contributing to the result. This could be further explored, as above, or at least acknowledged in the manuscript.

Yes, we agree with the reviewer. We discuss this limitation in the discussion section (pages 16-17) as it is difficult to adopt a “perfect method” for these issues.

2) Yes, as detailed the authors have discussed this, as above.

Thank you

3) The common Transcription Factors explanation detailed here does not appear to be included in the revised manuscript, unless I missed it? Furthermore, as the authors state “Our difficulty in replicating results in surrogate cell types after removal of underlying genetic variability is in keeping with observations from other diseases.” Therefore, that the previous positive blood tissue results, prior to Gap Hunting probe exclusion, were enriched for confounded genetic effects.

This is my second issue, in that these caveats should be directly declared. It should be stated in the paper that these surrogate tissue results were previously strongly confounded by genetic effects and that a minority of undetected genetic effects could still be present. Also, as well that common inflammatory cell changes could be contributing to the signal seen across both different tissue type samples.

We agree with the reviewer therefore we specifically included a statement “Multi tissue transcription factors could explain the replication of differentially methylated loci in multiple cell types. On the other hand, our difficulties in replicating results in surrogate cell types after removal of underlying genetic variability indicate that sequence variations could also drive cell type independent methylation changes which is in keeping with observations from other diseases⁵⁴⁻⁵⁷. Future studies shall aim to dissect the direct contribution of environmental factors and sequence variations in cytosine methylation in the kidney.” (pages 16-17).

In addition, to acknowledging the effect of cell type heterogeneity. “Despite all our efforts we cannot exclude the contribution of cell heterogeneity to the observed methylation changes.” (page 16)

4) Good to have explored the potential for ageing-related epigenetic changes to be contributing to this signal and excluding this possibility.

Thank you.

5) Additional details on the enrichment analysis have been provided.

Thank you.

6) Again, good to explore and define that these are explicitly kidney-specific enhancers.

Thank you.

7) Thanks to the authors for providing this additional detail regarding the expression analysis – and the use of the Shi et al. method is appropriate for this.

Thank you.

1. Zheng et al. (2018) Epigenomics 10.2217/epi-2018-0037

Reviewer #2 (Remarks to the Author):

Point 1. While it is appreciated that mechanisms underpinning DNA methylation might be beyond the scope of the study, the illumina 450k array allows the investigators to expand on the CG methylation from the signatures on the array to implicate genes and pathways that contribute to DKD. Defining the probes on the array is an important first step to the study and extending the analysis to identify hypermethylation and hypomethylation groups should identify genes containing differentially methylated sites in promoters or gene bodies using analyses such as IPA as recently shown by Chen et al (Proc Natl Acad Sci U S A. 2016 May 24;113(21):E3002-11).

Therefore, based on the original review points raised, understanding what methylation connections with regulatory pathways are not necessarily outside the scope of the current study. With this information the authors should be able to assign what methylation sites and states (reduced or increased or unchanged) can be used to improve predictive utility for DKD. This is an important consideration that was not addressed and should receive more attention for the readers of this journal. In addition, Chen et al were able to show the functional control of methylation using cells in experiments exposed to low-high glucose showing gene expression was inversely correlated with changes in DNA methylation. An important note here is the methylation datasets from both monocytes and whole blood cells by Chen are informative to the 115 blood samples derived from subjects examined in the replication cohort by Gluck et al. An assessment would have been informative.

Thank you for this suggestion. We have expanded our work beyond just simply reporting methylation differences.

1. **We report the magnitude and direction of methylation differences on Figure 1.**
2. **We report the nearest (and likely affected) gene on Tables 3 and 5.**
3. **We used human kidney ChIP-Seq data to functionally annotate the differentially methylated regions. We listed the functional annotation of these probes in Supplementary Tables 4 and 8. Probes on regulatory regions are more likely to be functionally important.**
4. **We have performed pathway analysis (suggested by the reviewer), using the Ingenuity Pathway analysis (IPA) and DAVID tools. These results are reported in Supplementary Table 5 and IPA analysis is shown in Supplementary Figure 4. For top probes that improve our progression model, DAVID analysis is shown in supplementary table 9 and IPA analysis is shown in Supplementary Figure 10.**
5. **To understand the potential functionality of the methylation changes, we examined the correlation between methylation and gene expression changes in the human kidney, by simultaneously analyzing gene expression and methylation changes in the same samples. We**

report loci where methylation changes in the kidney were associated with gene expression changes in the same samples (Table 3 and 5 and Supplementary Tables 6 and 11).

6. Finally, to show that methylation levels directly influence gene expression we have treated kidney tubule cells with Dnmt1 inhibitor 5AZA and analyzed methylation and gene expression changes. We focused on examining probes that were associated with the degree of interstitial fibrosis and the methylation of these probes already showed association with gene expression levels in the kidney (n=5) (Table 3). In addition, we examined probes that improved kidney function decline regression models and also showed association with gene expression in kidney samples (n=5) (Table 5). Total of 10 probes. Of these 10 probes 5 showed methylation changes after 5AZA treatment and 2 of the 5 probes (40%) also showed corresponding gene expression changes. For these sites, we were able to verify that methylation changes are directly associated with gene expression changes (i.e. this is not just as an association as we previously observed). Supplementary Figure 13 shows these results.

Point 2. Methylation in tubules and blood samples. The authors discuss this important pitfall in the discussion on page 15, however, readers of this journal need to see possible shared mechanisms that drive methylation in the multiple cell types. Describing what methylation states is only the beginning and the general feeling by the authors is this is beyond the study scope but should not be. Indeed, there are experiments that assess the functionality of methylation states and sites that are important in providing knowledge showing methylation is not an epiphenomenon. Chen et al provide examples of methylation changes in both WB and monocytes associated with gene expression changes.

To further examine the direct relationship between cytosine methylation and gene expression changes we performed an experiment with cultured proximal tubule cells (HKC8 cells). We artificially demethylated cytosines using the DNA methyltransferase inhibitor, 5-aza-2deoxycytidine. We measured cytosine methylation by Infinium 450K arrays and gene expression by AffymetrixST1.0 arrays.

We focused on examining probes that were associated with the degree of interstitial fibrosis and the methylation of these probes already showed association with gene expression levels in the kidney (n=5) (Table 3). In addition, we examined probes that improved kidney function decline regression models and also showed association with gene expression in kidney samples (n=5) (Table 5). Total of 10 probes. Of these 10 probes 5 showed methylation changes after 5AZA treatment and 2 of the 5 probes (40%) also showed corresponding gene expression changes. For these sites, we were able to verify that methylation changes are directly associated with gene expression changes. Box plots for methylation levels and cis gene expression levels are shown in Supplementary Figure 13. We therefore demonstrate the role of observed methylation differences in regulating gene expression changes. Understanding the upstream regulators of methylation remains a complex issue. We admire the elegant work of Chen et al. It is difficult to directly link high glucose, methylation and gene expression changes, given that hyperglycemia regulates a large number of pathways. In addition, there are multiple other factors in diabetes that could change the epigenome. For example, underlying genetic variation as strongly pointed out by reviewer 1 could also contribute to methylation changes.

For these reasons we limited our analysis to understand the correlation between methylation and gene expression changes.

Point 3. Given that small numbers of individuals are studied and present with varying degrees of DKD and the methylation probe cg21048700 is identified in a kidney inactive region which is thought to be "associated" with COL3A1. The link between the methylated probes and genes implicated with DKD is not strong and raises questions as to whether the methylation is beyond chance? Additional data is

necessary

to

address

this.

Thank you for this comment.

1. While our sample size is modest, it is still the largest for tissue-based methylation analysis. Indeed, we would like to point out again that this is the first tissue-based analysis for diabetic kidney disease. If our work can be published other scientists will be able to build on our dataset and expand the sample size and the field will grow.
2. We used the kidney function and methylation data as continuous variables to improve the power.
3. The relationship between the probes cg21048700 and COL3A1 are based on permutation, indicating that the relationship is significant compared with random chance. We have shown the strong association of this relationship in Supplementary Figure 11 The effect size of this relationship (beta) is strong -0.526 with a p-value of 1.34E-07.
4. The methylation of probe cg21048700 and COL3A1 expression level is strongly correlated with % Interstitial Fibrosis on histology as shown below:
5. We included the association of cg21048700 and COL3A1 based on reviewer #3's observation that this is in fact potentially an important pathway in the pathogenesis in kidney fibrosis. Others have shown that collagen turnover may be an important prognostic marker in for certain forms of kidney disease (Genovese et al NDT 2015;31 472-479).
6. While this probe does not colocalize with enhancer marks when a whole kidney (mostly tubule cells) was studied, it is localized to an enhancer region in human fibroblast (ENCODE data). We propose that this site might be important for regulating collagen3 expression in fibroblast, but probably not in kidney tubule cells. Since fibroblasts only represent just a small percentage of the whole kidney, whole tissue analysis is likely not sensitive enough.

Abstract: the contemporary use of the terms "genome wide cytosine methylation" is confusing when attached to 450K array and this will be noticed by the broad Nature Comm readership, furthermore, this is highlighted with the technological advances of the 850k array which many do not attach "genome wide cytosine methylation". Whilst it is understood that the 450K array is a cursory assessment of methylation, it by no means would be referred to as "genome wide" and this appears to be a misunderstood here. It is simpler and more accurate to use "Illumina Infinium 450K arrays".

Thank you for this suggestion, we changed the abstract. We used the term genome wide as a method that covers the whole genome (as opposed to site specific) we did not mean to imply that this is base resolution method. For example, genome wide association analysis (GWAS) is not performed at base resolution level, but the whole genome is covered (not site specific).

Methods/Design: The authors concede the replication cohort using blood samples is likely to be different from the kidney tissues and this could be the case for methylation. The article by Chen et al (Proc Natl Acad Sci U S A. 2016 May 24;113(21):E3002-11) show dramatic changes in DNA methylation including a range of genes and probes using the Illumina 450k array, the same method used in this article. Infact, that article is important, not only because of the relevant experimental design and 450 array approaches, but is relevant because they also identified probes from the 450k array associated with genes including TXNIP, CYB5b and many others. The dataset by Chen et al is rich with valuable information that could have strengthened the Gluck et al article. It is unclear why this relevant dataset that shows methylation in monocytes and whole blood was not carefully explored in relation to the current article. Any analysis and comparison would be useful, comparing similar technologies (450k array) and disease (diabetic complications).

Thank you.

Thank you, we have now included look-up analysis of the Chen et al data under Supplementary Table 12. We were able to confirm nominally significant changes in the Chen dataset.

We agree that the paper by Chen et al is an important milestone, however, we would like to point out that we used different statistical criteria and model adjustment for data analysis (FDR by Gluck and nominal p value by Chen). We used different tissue type (kidney vs. blood) and we used different cohorts (type1 vs. type2 diabetes). Additionally, we used different data analysis, for example we have tried to adjust for genetic variation by using GapHunter program. Despite all these significant differences we were able to validate some sites using nominal p-values indicating consistency.

Power calculations: 80% is too low for a study of the size and what appears to be replicated between blood and kidney tissue may very well reflect probe bias. This is why an independent method is useful.

Thank you. The power of the study depends on the effect size. The article by Tsai et (2015 Aug; 44(4): 1429–1441. Int J Epidemiology) includes power calculations estimations for varying effect sizes (or mean methylation differences) for Illumina 450K beadchips. They estimate that for case control studies approximately 25 pairs, will have 100% power to detect mean methylation difference 30%. They note that for cohorts like ours with increased case:control ratio (1:2 or 1:4) that there is increased power to detect smaller changes³. Most importantly, we did not analyze the data in a case control manner, but as continuous outcome, which improves the power.

In summary, small studies are well powered to detect large effect size differences. The best example of this is that the first GWAS study on macular degeneration has identified regions with genome wide significance, despite the very small sample size. Current GWAS studies using much larger sample sizes can identify changes with smaller effect sizes and regions identified by prior smaller studies tend to replicate in larger dataset.

This is the first study to report methylation changes in diabetic human kidney samples, so have used another kidney cohort to validate fibrosis associated as there is no other study available for validation. We fully agree that independent validation is useful and we performed such independent validation. We also would like to point out that no other prior studies have performed independent validation cohorts.

Furthermore, we would like to highlight that we used a blood sample replication cohorts to validate our observation.

Reviewer #3 (Remarks to the Author):

1. Patient groups are still poorly characterized, despite the authors' claim that they have provided a comprehensive clinical and histopathological information.

We are a bit surprised by this suggestion. In our view this is the first manuscript that uses the “gold standard” criteria for DKD description, which is histological diagnosis. We provide a detailed 19 point histopathological description of the cohort and a full description for the subjects and samples. Every other prior study and 99% of the published clinical literature use the presence of GFR decline or albuminuria in patients with diabetes. However, we do not know whether or not such subjects actually have diabetic kidney disease.

Additional comments from Reviewer #3:

Reviewer #3 (Remarks to the Author):

- As requested in the first revision, the authors should provide at least the following data: blood pressure, metabolic control, serum lipids, concomitant drug treatment such as ACE inhibitors, ARBs, HMGCoA inhibitors, blood glucose and lowering agents.

As stated above we are a bit surprised by this suggestion and would like to understand a strong rationale for this statement. In our view this is the first manuscript that uses the “gold standard” criteria for DKD description, which is histological diagnosis. We provide a detailed 19 point histopathological description of the cohort and a full description for the subjects and samples. Every other prior study and 99% of the published clinical literature use the presence of GFR decline or albuminuria in patients with diabetes. However, we do not know whether or not such subjects actually have diabetic kidney disease.

We would like to mention that we have provided the blood pressure information in the manuscript. We provide metabolic control such as HbA1c when available (not for non-diabetic subjects). Due to the sample size we cannot adjust for medication use and this information was only partially available and only for the time of sample collection. As the role of serum lipids and lipid lowering in DKD development is a bit controversial it is hard to make strong case for this information, which is a potential limitation of the study.

In particular, blood pressure data are of key importance. It is concerning that the authors adjusted for hypertension (yes/no), instead of using a continuous variable (e.g. MAP) in the reported regression models (see page 6, Figure 1). The above approach generates some questions that need to be addressed:

- 1. We would like to emphasize that we used machine learning methods (LASSO) to identify clinical, histological, gene expression and methylation variables to predict GFR decline. This is a key novelty of the work. We do not have control over the variables that predict kidney function decline in machine learning models. In our review of the literature no prior studies have used LASSO for model selection, as most prior models mostly picked variable “intuitively” or used a stepwise approach to model selection. It is also important to note that LASSO is a shrinkage model that penalizes for having too many variables in the model. It might be important to reanalyze some of the clinical observational cohorts using LASSO. For example, if a variable closely correlates with BP or HTN status LASSO will not use that variable.**

2. We would like to point out that the outcome in our study was different from prior publications. We examined GFR slope and interstitial fibrosis. In our view the role of BP has not been formally analyzed for these outcomes.
3. We acknowledge the sentiment on the role of hypertension in development of CKD. Our review of the literature indicates that while some studies have indicated the role of BP in progression others did not. For example, the study by Tangri et. al. JAMA 2011 found, that BP did not predict renal outcome. Further external validation of the Tangri et al equation in a very large cohort (JAMA 2016) show excellent discrimination of the formula and again show that BP did not add to the predictive accuracy for renal outcome. The blood pressure recommendation goals for patients with diabetes varies significantly between different organizations.
4. In our study, using a univariate analysis, BP did not correlate with our histological outcome (degree of tubulointerstitial fibrosis) (beta 0.147, p-value 0.22). However, HTN diagnosis did correlate with degree of interstitial fibrosis (beta 0.28, p-value 0.011); therefore, we used HTN as a covariate in our regression analysis. This may just reflect that a single blood pressure reading might not reflect long term blood pressure measurements or a residual effect of hypertension status despite medical management.
5. Similarly, as reported by prior studies, HTN did correlate with kidney function decline as outcome. In addition, BP (or MAP) at the time of kidney tissue procurement, poorly correlated (beta -0.07, p-value 0.57) with GFR decline. In our view, this may reflect the fact that medically treated HTN mitigates the effect of HTN on kidney function decline.
6. Finally, given the reviewer's concern we "forced" hypertension into the CKD progression model, and we identified 67 methylation probes that significantly improve the model (FDR<0.05) including 2 of the same top probes previously identified and associated with gene expression changes (EGF and HCLS1). We have included this result as Supplementary Table 10.

- The operational definition of 'hypertension' (e.g. DBP above 80 mmHg or use of antihypertensive agents) is not provided by the authors. From a clinical point of view, however, it is essential to clarify this aspect.

We clarified this issue. We used the hypertension diagnosis code as a definition of hypertension. In general, we think it is hard to find a correct definition for HTN as multiple organizations use different recommendations.

- Whatever definition is adopted by the authors, 'presence of hypertension' is determined on the basis of continuous BP values. It is therefore difficult to believe that the authors cannot avail themselves of original (continuous) data.

Please see response above. Please note that machine learning was used to identify factors for progression, but blood pressure was not picked by the algorithm. Please see results when blood pressure data was forced into the model.

- It is notable that for the independent validation cohort some regression models included mean arterial pressure (as a continuous variable) among the independent covariates, instead of 'hypertension'

(see page 12, Figure 5). Please explain why the authors didn't use the same independent variable in order to represent blood pressure.

We used published blood data as a replication cohort as kidney data was not available. This progression model was developed using prior knowledge and intuition and the authors included BP. In our manuscript we used a computer learning program LASSO to determine significant variables for the CKD progression model. As discussed above this model penalizes for adding too many variables into the model. In our analysis the inclusion of variables was unbiased, which is a key strength of the study.

Please see discussion above and review supplemental table 10 where we show data by forcing BP information into the multivariable model.

- The authors' choice is also suboptimal from a methodological point of view. Actually, it is always preferable to use continuous covariates instead of the same 'dichotomized' variable, because the former provides more accurate estimates.

Thank you for this interesting comment.

1. Please see discussion above. We used machine learning for variable selection.
2. Second, we are not aware of studies correlating BP with renal structural outcomes such as kidney fibrosis as outcome.
3. Third since BP treatment has a significant effect on reducing renal outcome, but may not completely eliminate all residual effects of HTN we think that there is an important biological basis to potentially use HTN as binary outcome.
4. Finally, to respond to this comment we “forced” blood pressure into our CKD progression model for the primary kidney cohort (results under supplemental table 10) and found that the majority of progression is explained by baseline GFR (40.1%) followed by diabetes (6.89%), then Age (4.4%) and finally HTN (1.5%) or Systolic BP (0.41%).

2. The authors claim that they have clarified the study design in the abstract. This is simply not true. Moreover, they did not provide any information about study setting, participating centers, recruitment period and data monitoring, as requested.

We would like to apologize. We clarified the study design in the rebuttal. Unfortunately, we cannot include all clinical information requested by the reviewer to the abstract due to word limitation so eventually it was deleted from the final submission.

We described the collection of the tissue samples and the clinical information in the methods section. We would like to point out that this is not a simple epidemiological dataset, the key aspect of the work is having tissue samples, detailed histological analysis and methylation data information and the use machine learning to identify variables that predict the different outcomes.

We have added the following description to the methods section: “The study used cross-sectional design. The samples were collected from Albert Einstein College of Medicine Montefiore Medical Center between the years of 2007-2011. Samples were de-identified and the corresponding clinical information available at the time of nephrectomy was collected by an honest broker. The study was approved by the Institutional Review Boards at the Albert Einstein College of Medicine Montefiore Medical Center (IRB 2002-202) and the University of Pennsylvania (IRB 815796).” These samples are considered medical discard and the samples were collected de-identified and the clinical data was added by an honest broker. For this reason, this is not considered a clinical trial and no DSMB was needed.

3. The authors asserted that in the primary cohort they included controls (non-diabetic CKD patients). However, this has not been specified in the manuscript, but, more importantly, no clinical data on diabetic and non-diabetic CKD patients (controls) have been provided. Data of the two patient populations should be shown. The choice of non-diabetic CKD patients is crucial, as they must have the same degree of renal insufficiency and proteinuria as diabetic patients.

We remain confused by these comments. Please see below response to additional comments below. The primary cohort did not include subjects with non-diabetic CKD. The study did not use CKD as outcome. We used continuous variables as outcome; structure damage and rate of kidney function decline.

- The authors never mentioned before that the primary cohort only had 22 diabetic CKD patients and it never emerged that the controls were healthy patients without any kind of kidney injury. Similarly, the replication kidney cohort should be described better, specifying how many patients are affected by diabetic CKD and how many by hypertensive CKD.

Thank you. We are very confused by statements of the reviewer. Diabetic CKD was not an outcome in this study and the controls were not healthy subjects without kidney injury.

The cohort was NOT analyzed in a case control manner as detailed in the manuscript. As highlighted by this reviewer many times continuous outcome analysis is more powerful than case control analysis. Indeed, the reviewer made 5 comments on continuous vs binary use of BP vs HTN. The primary cohort only had diabetic CKD patients (n=22). We used an external validation cohort to replicate the methylation and fibrosis association. This replication kidney cohort contained both diabetic CKD and non-diabetic CKD patients (diabetic CKD = 10, non-diabetic CKD = 38). The controls included patients with diabetes in the absence of kidney disease (n=11) as well as hypertension in the absence of kidney disease (n= 25). As stated in the manuscript we did not use DKD as an outcome. DKD is complex phenotype. Clinically both GFR decline and albuminuria describes DKD and these 2 definitions do not fully overlap. We used the gold standard histological definition. At present, we do not know the correlation between gold standard vs clinical DKD diagnosis. We replicated structural changes of DKD, in a cohort where the etiology of fibrosis was related to diabetes or hypertension.

We do not wish to present the data as a case control data as it was not analyzed as a case control dataset. We would like to point out that the CKD definition of eGFR<60 is highly arbitrary definition and differences between samples of eGFR 59 and 20 is highly significant while changes in samples between GFR of 61 (no CKD) and 59 (no CKD) is probably not greater than chance.

Cohort characteristics are shown in Table 1-2 and Supplementary Tables 1-3. The primary cohort did not include non-diabetic CKD patients. The primary kidney cohort only had diabetic CKD patients (n=22); the replication kidney cohort contained both diabetic CKD and non-diabetic CKD patients (n=48).

- In order to identify the best control, it is indispensable to have a clear idea of what one wants to demonstrate. Comparing CKD patients (diabetic plus hypertensive) to healthy patients, as the authors did, allows one to identify a methylation signature specific to CKD, and not to diabetic kidney disease. In this case, the authors should change their conclusions accordingly. By contrast, to define cytosine methylation changes specific to diabetic kidney disease, the authors should compare diabetic kidney disease patients with patients with the same degree of renal insufficiency and proteinuria as diabetic ones.

Thank you for the note. There seems to be a significant misunderstanding.

- 1. We would like to emphasize that this is NOT a case control study as suggested by the reviewer.**
- 2. The definition of CKD is having a GFR under 60ml/min/1.73m². We did not use this outcome. We do not understand the comment of the reviewer.**
- 3. We feel that figures 2 and 5 clearly described our analysis and outcomes. The goal is to understand DKD (hence the title). DKD however has multiple endophenotypes, such as eGFR, albuminuria, glomerulosclerosis, fibrosis and GFR decline. We have analyzed two phenotypes as continuous outcomes: 1) degree of interstitial fibrosis and 2) eGFR decline.**
- 4. In addition, it seems that the reviewer has misunderstood the analysis. The primary cohort was a cohort of subjects with varying degrees of renal insufficiency and kidney function decline. In this cohort, 41 subjects carried a diagnosis of DM and 65 carried a diagnosis of HTN. In this primary cohort, all subjects with GFR<60 had diabetes and evidence of DKD on histologic analysis (n=22). Subjects with GFR>60 included those with and without diabetes and with and without hypertension. As such, in our linear regression analysis, we adjusted for both DM and HTN and the above mentioned outcomes were used.**
- 5. For the structure changes in DKD, we used a cohort of subjects with diabetic and hypertensive fibrosis, given the significant overlap between histological manifestation of the tubulointerstitial changes. We would like to highlight that all models were adjusted for diabetes.**
- 6. Our CKD progression model was adjusted for baseline kidney function. Given that this is not a case control design and we adjusted the data for these covariates, we do not see the value of matching the data.**

4. I recognize the step forward upon filtering out with Gap Hunter method. However, this new approach has not confirmed some probes associated with interstitial fibrosis found in the original paper's analysis, in particular as regards the one correlated with the stem cell transcription factor HOPX (Hop homeobox) levels that, according to the authors, could have a pivotal role in disease development.

There seems to be a misunderstanding.

- 1. The observed methylation changes around HOPX are correct.**
- 2. Upon the suggestion of reviewer1 we removed probes where the methylation change in the vicinity of sequence variations, such as the HOPX region.**
- 3. The argument here is not about whether or not the methylation difference exists, but about the cause of the observed methylation change (environmentally induced vs the result of genetic variation).**
- 4. We will need to generate genotype information for these exact samples to understand the cause of methylation changes in the HOPX region.**

The authors didn't specify whether the degree of cg24818418 methylation associated with EGF (Epidermal Growth Factor) transcript level, and able to predict kidney function decline, was validated in the independent cohort. This should be the most important confirmation finding of their study.

We are a bit confused by this comment as this information is included in the manuscript.

- 1. We show that the methylation of EGF probe predicts eGFR decline (Fig4).**
- 2. The EGF probe is on a kidney specific regulatory region and the methylation is associated with gene expression changes.**
- 3. We dedicated an entire figure (Figure 4) to show this important finding of the study.**

4. We show the effect and p-value (beta 0.008, p-value 0.049) of this probe with degree of interstitial fibrosis for our multivariate analysis in Supplementary Table 11.
5. On univariate analysis, the relationship is strong (beta 0.46, p-value 1.2 E-05). Furthermore, transcript level of EGF is well correlated with degree of interstitial fibrosis on univariate analysis (beta -0.50, p-value 4.54E-06).
6. The association between fibrosis and EGF transcript level has also been validated by Beckerman et al (eBioMedicine 2017). The association between EGF transcript and fibrosis and EGF and kidney function decline has been observed by Ju et al (Sci Trans Med 2016).

6. The authors did not provide a convincing answer about the use of surrogate cells to define epigenetic changes observed in the kidney. Two of three papers cited by the authors in their defense actually do not support their claims. Indeed, Hannon et al. found that for the majority of DNA methylation sites, interindividual variation in whole blood is not a strong predictor of interindividual variation in the brain, while Klein and co-authors asserted that “it is unlikely that there is a strong, direct correlation between the epigenomes of the brain and blood such that measuring a given epigenomic feature in blood offers a reasonable proxy for the same feature in the brain for most genomic loci”.

This is a good point, however, we would like to point out that the 3 reviewers seem to have differing views on this issue. The goal of this study is to describe methylation changes in human kidney samples of subjects with diabetic kidney disease, as the kidney tubule is one important cell type for diabetic kidney disease development.

The entire field of epigenetics does not have sufficient information to come to a solid conclusion on the use of surrogate cell types for epigenome wide association studies. In the study performed by Hannon et al (“Interindividual methylomic variation across blood, cortex, and cerebellum: implications for epigenetic studies of neurological and neuropsychiatric phenotypes”), 1.19% of whole blood variable methylation probes were “strongly” and 3.68% were “moderately” correlated with changes in the cerebellum. The authors postulate the role of underlying genetic variation and allele specific DNA methylation as a potential explanation for these similarities and reference the works of other smaller studies with similar comparisons (for example, Sliker et al “Identification and systematic annotation of tissue-specific differentially methylated regions using the Illumina 450k array”). Surrogate cell types are often more accessible than the tissue of interest (as in the case with our study) and markers that are able to be validated in the blood may be of increased clinical utility which is why we strove for this comparison.

Overall the use and role of surrogate cell types for epigenetic studies is not a solved issue. Making it even more important to analyze epigenetic changes in the cell type of interest. If possible, we would like to avoid making strong statements on the use of surrogate cell types for epigenetic studies.

7. The authors should properly cite the method they used to estimate the power of their sample size (Tsai et al., ?) in the main text.

We corrected this issue.

Minor points:

- Supplementary tables are erroneously named. Please check and correct.
- The authors should re-write the sentence on page 8, lines 14-17 “Fold change...analysis”.

Thank you. We corrected our labeling error.

We re-wrote this to read: “Compared to a random selection of probes our set of 65 probes (that are associated with DKD and fibrosis) showed a 4.5-fold enrichment to be localized to a kidney enhancer region, suggesting their functional importance in the kidney.”

REVIEWERS' COMMENTS:

Reviewer #1 (Remarks to the Author):

All my concerns have now been replied to - Thanks

Reviewer #2 (Remarks to the Author):

Contrary to the popular belief that DNA methylation is inversely correlated with expression in disease, this article in its current form indicates that understanding the role of DNA methylation in kidney disease is complicated. Additional experiments and new data included in the revised work while helpful raise more questions regarding the relevance of differentially methylated genes in diabetic kidney disease. Let's consider the new data included in Tables 3 and 5. The new data is a worthwhile addition to the article, but the list of replicated probes is all too brief and difficult to understand why the authors have only presented a shortened list. However, the main criticism of the data is discussed below.

New Table 3: a brief list of 5 methylated probes derived for the Illumina 450K array show the position of the methylated CG site to the nearest transcription start site of cis-acting genes are a long distance away and more than 10k base pairs (for example IFI16 and CCND2) with the majority of CG sites located 137k (AIM2) 279k (FCER1A)..., bp away from the TSS. Based on the CG locations that are 137,045 or 279,663 bp away from the nearest gene (AIM2, FCER1A , etc) it remains difficult to understand how these distances and CG sites are directly regulating expression of predicted genes that are implicated to be functionally relevant to interstitial fibrosis.

New Table 5: the same critique described above applies to the list of probes that improve model of kidney function decline. These results are neither described on pages 8-9 for table 3 or page 11 for table 5 in the results sections and surprisingly the issue of CG site distance to the nearest TSS is neither interpreted nor is it described as a limitation in the discussion. The issue of functional CG methylation is important, and it is difficult to imagine without direct experimental evidence that a CG methylation site 10,000, 100,000 or 400,000 base pairs away from the nearest TSS of a cis-acting gene is regulating gene expression.

The experiments using 5AZA in HKC8 cells to show functional relevance and methylation mediated gene regulation are interesting but problematic and flawed. Of the ten probes on the array 5 showed methylation changes after 5AZA and 2 of the 5 probes were correlated with gene expression. A close look at the data shows confusion understanding the role of DNA methylation and the complex interpretive issues with the dataset. The title to SF13 describes CG methylation of two probes linked with expression of genes implicated in interstitial fibrosis, specifically, C1S and HCLS1, however, HCLS1 is listed in Table 5 as a probe that improve model of kidney function. While this might be a minor point the next issue is more complicated and relates to the methylation probe of the C1S gene is 164,195 bp away and for HCLS1 is 295,485 bp away from their respective transcription start sites. Genic methylation occurring at a distance from the TSS is positively correlated with gene expression (some examples of these published in Nature, 466 (2010), pp. 253-257 and Genome Res., 23 (2013), pp. 555-567) which is not to be confused with the DNA methylation found in promoters that is associated with gene repression. The authors rightly or wrongly expect 5AZA to cause hypomethylation in genic regions in the same way as promoters are demethylated by 5AZA and subsequently causes gene reactivation. Gene body methylation may not necessarily operate in the same way as promoter methylation and this remains a subtle but important difference between promoter and gene body DNA methylation (Cancer Cell 26: 4, 13 Oct2014, Pages 577-590). There is a confusion with the interpretation of the data and the authors have not considered this subtle but important difference nor is the current literature discussed. The implication that 5AZA causes demethylation of a CG methylation probe that is 164,195 bp away from C1S and 295,485 bp away from HCLS1 which are considered genic methylation sites is flawed and paradoxical because gene body remethylation is correlated with gene expression.

Based on these comments the role of methylation in DKD is unclear.

Reviewer #3 (Remarks to the Author):

The author's answers have clarified several aspects that have been neglected in the manuscript. The authors should improve their manuscript so that it will be more reader-friendly.

R- As requested in the first revision, the authors should provide at least the following data: blood pressure, metabolic control, serum lipids, concomitant drug treatment such as ACE inhibitors, ARBs, HMGCoA inhibitors, blood glucose and lowering agents.

A- As stated above we are a bit surprised by this suggestion and would like to understand a strong rationale for this statement. In our view this is the first manuscript that uses the "gold standard"

criteria for DKD description, which is histological diagnosis. We provide a detailed 19 point histopathological description of the cohort and a full description for the subjects and samples.

Every other prior study and 99% of the published clinical literature use the presence of GFR decline or albuminuria in patients with diabetes. However, we do not know whether or not such subjects actually have diabetic kidney disease.

We would like to mention that we have provided the blood pressure information in the manuscript. We provide metabolic control such as HbgA1c when available (not for non-diabetic subjects). Due to the sample size we cannot adjust for medication use and this information was only partially available and only for the time of sample collection. As the role of serum lipids and lipid lowering in DKD development is a bit controversial it is hard to make strong case for this information, which is a potential limitation of the study.

The problem of only partially available clinical information at the time of sample collection as far as of the potentially reduced generalizability of the selected patient sample should be mentioned in the text among the limitations of the study.

R- In particular, blood pressure data are of key importance. It is concerning that the authors adjusted for hypertension (yes/no), instead of using a continuous variable (e.g. MAP) in the reported regression models (see page 6, Figure 1). The above approach generates some questions that need to be addressed:

A- We would like to emphasize that we used machine learning methods (LASSO) to identify clinical, histological, gene expression and methylation variables to predict GFR decline. This is a key novelty of the work. We do not have control over the variables that predict kidney function decline in machine learning models. In our review of the literature no prior studies have used LASSO for model selection, as most prior models mostly picked variable “intuitively” or used a stepwise approach to model selection. It is also important to note that LASSO is a shrinkage model that penalizes for having too many variables in the model. It might be important to reanalyze some of the clinical observational cohorts using LASSO. For example, if a variable closely correlates with BP or HTN status LASSO will not use that variable.

We would like to point out that the outcome in our study was different from prior publications. We examined GFR slope and interstitial fibrosis. In our view the role of BP has not been formally analyzed for these outcomes.

We acknowledge the sentiment on the role of hypertension in development of CKD. Our review of the literature indicates that while some studies have indicated the role of BP in progression others did not. For example, the study by Tangri et. al. JAMA 2011 found, that BP did not predict renal outcome. Further external validation of the Tangri et al equation in a very large cohort (JAMA 2016) show excellent discrimination of the formula and again show that BP did not add to the predictive accuracy for renal outcome. The blood pressure recommendation goals for patients with diabetes varies significantly between different organizations.

In our study, using a univariate analysis, BP did not correlate with our histological outcome (degree of tubulointerstitial fibrosis) (beta 0.147, p-value 0.22). However, HTN diagnosis did correlate with degree of interstitial fibrosis (beta 0.28, p-value 0.011); therefore, we used HTN as a covariate in our regression analysis. This may just reflect that a single blood pressure reading might not reflect long term blood pressure measurements or a residual effect of hypertension status despite medical management.

Including in the Supplemental Material the above findings arising from the univariate analyses may help in better understanding the variables' selection process.

Similarly, as reported by prior studies, HTN did correlate with kidney function decline as outcome. In addition, BP (or MAP) at the time of kidney tissue procurement, poorly correlated (beta -0.07, p-value 0.57) with GFR decline. In our view, this may reflect the fact that medically treated HTN mitigates the effect of HTN on kidney function decline.

The lack of correlation between blood pressure and eGFR decline is clinically interesting and it should be added in the Supplemental Material. Moreover the Authors should recognize in the Discussion section the use of estimated GFR instead of measured GFR may be suboptimal in representing renal function in diabetes.

Moreover the authors should include the following information that are important and clinically relevant:

- in the abstract: primary cohort had diabetic CKD patients (n=22);
- in the method section: replication kidney cohort had both diabetic CKD and non-diabetic CKD patients (diabetic CKD = 10, non-diabetic CKD = 38). The controls included patients with diabetes in the absence of kidney disease (n=11) as well as hypertension in the absence of kidney disease (n=25).

REVIEWERS' COMMENTS:

Reviewer #1 (Remarks to the Author):

All my concerns have now been replied to - Thanks

Thank you.

Reviewer #2 (Remarks to the Author):

Contrary to the popular belief that DNA methylation is inversely correlated with expression in disease, this article in its current form indicates that understanding the role of DNA methylation in kidney disease is complicated. Additional experiments and new data included in the revised work while helpful raise more questions regarding the relevance of differentially methylated genes in diabetic kidney disease. Let's consider the new data included in Tables 3 and 5. The new data is a worthwhile addition to the article, but the list of replicated probes is all too brief and difficult to understand why the authors have only presented a shortened list. However, the main criticism of the data is discussed below.

Thank you for this comment. We have provided the full lists of methylation probes associated with outcomes in our supplementary information and supplementary data. Unfortunately, this information is too long to include in the main manuscript.

New Table 3: a brief list of 5 methylated probes derived for the Illumina 450K array show the position of the methylated CG site to the nearest transcription start site of cis-acting genes are a long distance away and more than 10k base pairs (for example IFI16 and CCND2) with the majority of CG sites located 137k (AIM2) 279k (FCER1A)..., bp away from the TSS. Based on the CG locations that are 137,045 or 279,663 bp away from the nearest gene (AIM2, FCER1A , etc) it remains difficult to understand how these distances and CG sites are directly regulating expression of predicted genes that are implicated to be functionally relevant to interstitial fibrosis.

New Table 5: the same critique described above applies to the list of probes that improve model of kidney function decline. These results are neither described on pages 8-9 for table 3 or page 11 for table 5 in the results sections and surprisingly the issue of CG site distance to the nearest TSS is neither interpreted nor is it described as a limitation in the discussion. The issue of functional CG methylation is important, and it is difficult to imagine without direct experimental evidence that a CG methylation site 10,000, 100,000 or 400,000 base pairs away from the nearest TSS of a cis-acting gene is regulating gene expression.

Thank you. We have added this important point to our results and discussion sections. We have included a caveat that functional link would require experimental validation given the large distances between some methylation probes and the postulated target

genes. We have included a brief discussion about mechanisms of epigenetic regulation as well.

The experiments using 5AZA in HKC8 cells to show functional relevance and methylation mediated gene regulation are interesting but problematic and flawed. Of the ten probes on the array 5 showed methylation changes after 5AZA and 2 of the 5 probes were correlated with gene expression. A close look at the data shows confusion understanding the role of DNA methylation and the complex interpretive issues with the dataset. The title to SF13 describes CG methylation of two probes linked with expression of genes implicated in interstitial fibrosis, specifically, C1S and HCLS1, however, HCLS1 is listed in Table 5 as a probe that improve model of kidney function. While this might be a minor point the next issue is more complicated and relates to the methylation probe of the C1S gene is 164,195 bp away and for HCLS1 is 295,485 bp away from their respective transcription start sites. Genic methylation occurring at a distance from the TSS is positively correlated with gene expression (some examples of these published in Nature, 466 (2010), pp. 253-257 and Genome Res., 23 (2013), pp. 555-567) which is not to be confused with the DNA methylation found in promoters that is associated with gene repression. The authors rightly or wrongly expect 5AZA to cause hypomethylation in genic regions in the same way as promoters are demethylated by 5AZA and subsequently causes gene reactivation. Gene body methylation may not necessarily operate in the same way as promoter methylation and this remains a subtle but important difference between promoter and gene body DNA methylation (Cancer Cell 26: 4, 13 Oct2014, Pages 577-590). There is a confusion with the interpretation of the data and the authors have not considered this subtle but important difference nor is the current literature discussed. The implication that 5AZA causes demethylation of a CG methylation probe that is 164,195 bp away from C1S and 295,485 bp away from HCLS1 which are considered genic methylation sites is flawed and paradoxical because gene body remethylation is correlated with gene expression.

Thank you for this comment. Based on this comment and at the suggestion of the editorial board, we have removed this experimental data from the finalized manuscript.

Based on these comments the role of methylation in DKD is unclear.

Reviewer #3 (Remarks to the Author):

The author's answers have clarified several aspects that have been neglected in the manuscript. The authors should improve their manuscript so that it will be more reader-friendly.

R- As requested in the first revision, the authors should provide at least the following data: blood pressure, metabolic control, serum lipids, concomitant drug treatment such as ACE inhibitors, ARBs, HMGCoA inhibitors, blood glucose and lowering agents.

A- As stated above we are a bit surprised by this suggestion and would like to understand a strong rationale for this statement. In our view this is the first manuscript that uses the “gold standard” criteria for DKD description, which is histological diagnosis. We provide a detailed 19 point histopathological description of the cohort and a full description for the subjects and samples.

Every other prior study and 99% of the published clinical literature use the presence of GFR decline or albuminuria in patients with diabetes. However, we do not know whether or not such subjects actually have diabetic kidney disease.

We would like to mention that we have provided the blood pressure information in the manuscript. We provide metabolic control such as HbA1c when available (not for non-diabetic subjects). Due to the sample size we cannot adjust for medication use and this information was only partially available and only for the time of sample collection. As the role of serum lipids and lipid lowering in DKD development is a bit controversial it is hard to make strong case for this information, which is a potential limitation of the study.

The problem of only partially available clinical information at the time of sample collection as far as of the potentially reduced generalizability of the selected patient sample should be mentioned in the text among the limitations of the study.

Thank you. We added this limitation in the discussion section.

R- In particular, blood pressure data are of key importance. It is concerning that the authors adjusted for hypertension (yes/no), instead of using a continuous variable (e.g. MAP) in the reported regression models (see page 6, Figure 1). The above approach generates some questions that need to be addressed:

A- We would like to emphasize that we used machine learning methods (LASSO) to identify clinical, histological, gene expression and methylation variables to predict GFR decline. This is a key novelty of the work. We do not have control over the variables that predict kidney function decline in machine learning models. In our review of the literature no prior studies have used LASSO for model selection, as most prior models mostly picked variable “intuitively” or used a stepwise approach to model selection. It is also important to note that LASSO is a shrinkage model that penalizes for having too many variables in the model. It might be important to reanalyze some of the clinical observational cohorts using LASSO. For example, if a variable closely correlates with BP or HTN status LASSO will not use that variable.

We would like to point out that the outcome in our study was different from prior publications. We examined GFR slope and interstitial fibrosis. In our view the role of BP has not been formally analyzed for these outcomes.

We acknowledge the sentiment on the role of hypertension in development of CKD. Our review of the literature indicates that while some studies have indicated the role of BP in

progression others did not. For example, the study by Tangri et. al. JAMA 2011 found, that BP did not predict renal outcome. Further external validation of the Tangri et al equation in a very large cohort (JAMA 2016) show excellent discrimination of the formula and again show that BP did not add to the predictive accuracy for renal outcome. The blood pressure recommendation goals for patients with diabetes varies significantly between different organizations.

In our study, using a univariate analysis, BP did not correlate with our histological outcome (degree of tubulointerstitial fibrosis) (beta 0.147, p-value 0.22). However, HTN diagnosis did correlate with degree of interstitial fibrosis (beta 0.28, p-value 0.011); therefore, we used HTN as a covariate in our regression analysis. This may just reflect that a single blood pressure reading might not reflect long term blood pressure measurements or a residual effect of hypertension status despite medical management.

Including in the Supplemental Material the above findings arising from the univariate analyses may help in better understanding the variables' selection process.

Thank you. We have added the univariate analyses to the Supplementary Information.

Similarly, as reported by prior studies, HTN did correlate with kidney function decline as outcome. In addition, BP (or MAP) at the time of kidney tissue procurement, poorly correlated (beta -0.07, p-value 0.57) with GFR decline. In our view, this may reflect the fact that medically treated HTN mitigates the effect of HTN on kidney function decline.

The lack of correlation between blood pressure and eGFR decline is clinically interesting and it should be added in the Supplemental Material. Moreover the Authors should recognize in the Discussion section the use of estimated GFR instead of measured GFR may be suboptimal in representing renal function in diabetes.

Thank you. We have added the correlation between blood pressure and eGFR decline to the Supplementary Information.

We have added to the methods section that we used estimated GFR using the Chronic Kidney Disease Epidemiology Collaboration (CKD-EPI) equation. While many research studies use measured GFR, several longitudinal studies have shown that use of estimated GFR correlates with hard outcomes such as mortality and kidney failure².

Moreover the authors should include the following information that are important and clinically relevant:

- in the abstract: primary cohort had diabetic CKD patients (n=22);
- in the method section: replication kidney cohort had both diabetic CKD and non-diabetic CKD patients (diabetic CKD = 10, non-diabetic CKD = 38). The controls included patients with diabetes in the absence of kidney disease (n=11) as well as

hypertension in the absence of kidney disease (n= 25).

Unfortunately, we are unable to add this information to the abstract since we were required to shorten our abstract significantly.

We have added to the methods section the clarifications about the primary and replication cohorts.

References

- 1 Kent, W. J. *et al.* The human genome browser at UCSC. *Genome Res* **12**, 996-1006, doi:10.1101/gr.229102 (2002).
- 2 Hsu, C. Y. & Bansal, N. Measured GFR as "gold standard"--all that glitters is not gold? *Clin J Am Soc Nephrol* **6**, 1813-1814, doi:10.2215/CJN.06040611 (2011).